# Integration of overlapping sequences emerges with consolidation through medial prefrontal cortex neural ensembles and hippocampal–cortical connectivity

**Alexa Tompary[1]\*, Lila Davachi[2]**

[1]Drexel University, Philadelphia, United States; [2]Columbia University, New York, United States

**Abstract** Systems consolidation theories propose two mechanisms that enable the behavioral integration of related memories: coordinated reactivation between hippocampus and cortex, and the emergence of cortical traces that reflect overlap across memories. However, there is limited empirical evidence that links these mechanisms to the emergence of behavioral integration over time. In two experiments, participants implicitly encoded sequences of objects with overlapping structure. Assessment of behavioral integration showed that response times during a recognition task reflected behavioral priming between objects that never occurred together in time but belonged to overlapping sequences. This priming was consolidation-dependent and only emerged for sequences learned 24 hr prior to the test. Critically, behavioral integration was related to changes in neural pattern similarity in the medial prefrontal cortex and increases in post-learning rest connectivity between the posterior hippocampus and lateral occipital cortex. These findings suggest that memories with a shared predictive structure become behaviorally integrated through a consolidation-related restructuring of the learned sequences, providing insight into the relationship between different consolidation mechanisms that support behavioral integration.

**\*For correspondence:**
at3549@drexel.edu

## Editor's evaluation

This important study investigates how memory representations are transformed over time (24h period), using a novel behavioral task and fMRI. The work advances our understanding of the neural processes supporting the behavioral integration of memories for distinct events that are never experienced together in time but are linked by shared predictive cues. Evidence supporting the claims is convincing, with inclusion of important control analyses that lend support to the authors' interpretation of the data.

## Introduction

There are now abundant demonstrations that after periods involving consolidation, episodic memories undergo transformations that allow individuals to integrate across overlapping experiences. Examples of enhanced behavioral integration with consolidation include transitive inference (*Ellenbogen et al., 2007*; *Lau et al., 2010*; *Werchan and Gómez, 2013*), the extraction of statistical regularities (*Wagner et al., 2004*; *Durrant et al., 2011*; *Durrant et al., 2011*; *Sweegers et al., 2014*; *Batterink and Paller, 2017*), category learning (*Djonlagic et al., 2009*; *Graveline and Wamsley, 2017*), and more (for

reviews, see *Chatburn et al., 2014*; *Lerner and Gluck, 2019*). What neural transformations support such consolidation-dependent integration across experiences?

Theories of systems-level consolidation posit that memories initially supported by the hippocampus become distributed across the cortex, through ongoing, coordinated reactivation of prior experiences (*Nadel et al., 2000*; *Squire et al., 1984*). Through this reactivation, the memory traces that are ultimately learned by cortex are thought to be highly structured and integrated representations built up from many overlapping experiences (*McClelland et al., 1995*). One class of theories posits that *neural* integration is accompanied by a psychological transformation such that memories supported by cortex are more gist-like and reflect the shared aspects across multiple events (Trace Transformation Theory (TTT); *Sekeres et al., 2018a*; *Winocur et al., 2010*). Thus, systems-level consolidation theories point to two neural mechanisms that could support such time-dependent *behavioral* integration of overlapping memories: (1) coordinated reactivation of memory traces between the hippocampus and cortex after learning, and (2) the emergence of cortical memory traces that reflect shared content across memories for different but overlapping events. However, evidence for this is limited and, thus, it is unclear whether and how these two mechanisms may jointly support the behavioral integration of overlapping experiences. Here, we present a new behavioral protocol to probe the implicit integration of events with overlapping content, and we use this protocol to investigate how cortical similarity and post-learning hippocampal–cortical coupling interact in supporting consolidation-dependent behavioral integration in humans.

## Integration and cortical similarity

According to TTT, the *behavioral* integration of related memories should be supported by the *neural* integration of cortical memory traces that reflects their overlap. Previous work has already pointed to medial prefrontal cortex (mPFC) as a key cortical region supporting integrated memories that emerge without the aid of consolidation. For instance, damage to this region impairs key memory integration behaviors like associative inference in humans (*Koscik and Tranel, 2012*; *Spalding et al., 2018*; *Warren et al., 2014*) and transitive inference in rodents (*DeVito et al., 2010*). Furthermore, activation of this region and its connectivity with the hippocampus increases when encoding episodes containing stimuli that overlap with recently learned information (*Kuhl et al., 2010*; *Schlichting and Preston, 2016*; *Zeithamova et al., 2012*), providing convincing evidence of its role in memory integration. Of particular relevance to the current study, patterns of activity in mPFC become more correlated for events linked through overlapping elements (*Schlichting et al., 2015*; *Milivojevic et al., 2015*) and also carry information about the common structure across many learned relationships (*Morton et al., 2020*; *Schuck et al., 2016*). With converging evidence spanning multiple species and methods is clear that mPFC supports the acquisition and immediate expression of integrated memories.

However, there is much less evidence in humans that the neural representations in mPFC are predictive of behavioral measures of memory integration over the course of systems-level consolidation, despite such strong behavioral evidence of consolidation-dependent behavioral integration (e.g. *Chatburn et al., 2014*) and the increasing involvement of mPFC in memory retrieval over time (*Bonnici et al., 2012*; *Frankland et al., 2004*; *Takashima et al., 2006*; *Takashima et al., 2009*; *Takehara-Nishiuchi and McNaughton, 2008*; *Woodard et al., 2007*). In other words, a direct link between the two is lacking. There are some clues that such a relationship should exist. First, in rodents, neural ensembles in mPFC become more selective for common features across two different associative memories and less selective to features unique to each (*Morrissey et al., 2017*). Second, there is evidence of delay-dependent changes in the content of memories in human mPFC (*Sekeres et al., 2018b*; *Krenz et al., 2023*). Third, there is evidence for neural integration of related memories in this region. Previously, we used a multi-variate pattern similarity approach to demonstrate that after a week, neural patterns during the retrieval of overlapping memories grew more correlated (*Tompary and Davachi, 2017*). *Audrain and McAndrews, 2022* observed a similar increase in correlation of overlapping memories over time in mPFC, although with notable differences in their protocol and consequent findings (see Discussion). However, as neither group measured the *behavioral* integration across those overlapping memories, the link between these time-dependent neural transformations and behavioral integration remains unknown. Thus, despite its theorized role in the expression of behaviorally integrated memories over time (*Preston and Eichenbaum, 2013*), to our knowledge,

no other human research has examined how consolidation may organize memory traces in mPFC to support their behavioral integration.

## Integration and hippocampal–cortical coupling

Another fundamental component of systems-level consolidation theories is the coordinated reactivation of memory traces between the hippocampus and cortex after learning. One way to operationalize such post-learning coordination is by measuring changes in the the correlation of their blood oxygenation level dependent (BOLD) signals during resting-state functional magnetic resonance imaging (fMRI) after encoding new memoranda (*Tambini et al., 2010*; *van Kesteren et al., 2010*; *Tompary et al., 2015*; *de Voogd et al., 2016*; *Gruber et al., 2016*; *Schlichting and Preston, 2016*; *Murty et al., 2017*; *Liu et al., 2018*; *Tambini and D'Esposito, 2020*; *Audrain and McAndrews, 2022*). While hippocampal–cortical coupling is not a direct measure of the reactivation of specific memories, memory-related patterns of coupling are specific to category-selective cortical regions (*de Voogd et al., 2016*; *Murty et al., 2017*; *Vilberg and Davachi, 2013*) and emerge only in post-learning rest periods rather than pre-learning ones, making rest connectivity a likely signature of coordinated reactivation across hippocampus and cortex (*Tambini and Davachi, 2019*). The vast majority of studies investigating post-learning rest connectivity find that the magnitude of experience-dependent change relates to subsequent memory accuracy, with the exception of one demonstration that post-learning connectivity relates to explicit associative inference (*Schlichting and Preston, 2016*). To our knowledge, this is the only finding that relates post-learning rest connectivity to a measure of behavioral integration rather than memory accuracy of learned items, but it does not examine consolidation-related changes as the integration test was administered a few minutes after learning. Furthermore, while a handful of studies reveal relationships between post-learning rest connectivity and the neural integration of overlapping memories (*Tompary and Davachi, 2017*; *Audrain and McAndrews, 2022*), none to our knowledge has examined how these neural measures, jointly or separately, support behavioral integration. This leaves open the fundamental question of whether post-learning coupling supports the neural *transformation* of memories that renders them more integrated with other, similar memories, as predicted by TTT.

Which cortical areas, and which regions of the hippocampus, might coordinate their processing to support behavioral integration? Beginning with cortex, the earliest demonstration of post-learning connectivity implicate the involvement of category-selective cortex (*Tambini et al., 2010*). Numerous findings since then have shown that post-learning cortical connectivity with the hippocampus is governed by the category of the encoded memoranda (*Collins and Dickerson, 2019*; *Keller and Just, 2016*; *Murty et al., 2017*; *Schlichting and Preston, 2014*; *Vilberg and Davachi, 2013*). Because our stimuli are common objects, we focused on lateral occipital cortex (LOC), a region causally linked to the formation of object memories (*Tambini and D'Esposito, 2020*). Surprisingly, while there is ample evidence that hippocampal interactions with mPFC support integrative encoding (*Zeithamova et al., 2012*) and retrieval of remote memories (*Takashima et al., 2007*; *McCormick et al., 2015*), there is less evidence that hippocampal coupling with mPFC in post-learning periods leads to strengthening or integration of new memories (*Tambini et al., 2010*; although cf. *Bein et al., 2014*). More commonly, post-learning hippocampal coupling with mPFC is related to updating and integrating of memories that were either previously strongly learned, older, or consistent with prior knowledge (e.g. *van Kesteren et al., 2010*; *Schlichting and Preston, 2016*; *Liu et al., 2017*; *Cowan et al., 2020*; *Audrain and McAndrews, 2022*). Because we were interested in examining neural processes related to the integration of newly learned information, we chose to focus on hippocampal interactions with LOC over mPFC.

Turning to the hippocampus, while the bulk of post-learning connectivity investigations treat the hippocampus as a unitary structure, there is some work that investigates differential connectivity with cortical regions along its long axis. Here, there are observations that post-learning connectivity between posterior hippocampus and category-selective cortical regions relate to memory for objects and scenes (*Murty et al., 2017*; *Tambini et al., 2010*), whereas connectivity between anterior hippocampus and fusiform face area (FFA) has been shown to relate to memory for faces (*Liu et al., 2018*) and to explicit associative inferences across stimuli including faces (*Schlichting and Preston, 2016*). Connectivity with anterior or posterior hippocampus also depends on whether memory is rewarded, with memory performance for low-reward items correlated with connectivity between posterior

hippocampus and category-selective cortex and memory for high-reward items related to connectivity between anterior hippocampus and category-selective cortex (*Murty et al., 2017*). Taken together, the factors that may explain differential connectivity patterns along the long axis of the hippocampus remain difficult to tease apart: possibilities include the form of behavior tested (memory for discrete events versus integration across them), the cortical regions selective to the content of encoded material, or the intrinsic or extrinsic motivation for consolidation. The present study employed object stimuli and did not implement a reward manipulation, so we chose to focus on posterior hippocampus, mirroring past experiments (*Murty et al., 2017*; *Tambini et al., 2010*). However, it is important to note that we based this focus solely on offline connectivity findings. When considering how the long axis of the hippocampus differentially supports memory integration, the anterior hippocampus is most often implicated, albeit during more immediate or online integration of stimuli (*Schapiro et al., 2012*; *Schlichting et al., 2015*; *Ritchey et al., 2015*). Given these differences in function along the long axis, our choice of focusing on posterior hippocampus was exploratory.

### Integration of sequential regularities

Rather than employing a more traditional explicit inference task, we generated sequences ending in overlapping and distinct objects (i.e. paradigmatic relations; *Luo and Zhao, 2018*; *McNeill, 1963*). We made this decision for several reasons: (1) the learning of statistical regularities relies on the hippocampus and benefits from consolidation processes (*Durrant et al., 2011*; *Durrant et al., 2013*; *Turk-Browne et al., 2009*; *Schapiro et al., 2012*; *Schapiro et al., 2014*), in line with predictions from TTT; (2) a statistical learning protocol enabled us to develop a behavioral task that relied on response times (RTs). In addition to statistical learning, RTs have been shown to reflect consolidation-dependent memory integration across several domains, including lexical integration of novel word learning (*Bowers et al., 2005*; *Davis et al., 2009*; *Coutanche and Thompson-Schill, 2014*), motor learning (*Kuriyama et al., 2004*; *Fischer et al., 2006*), and category learning (*Hennies et al., 2014*), making them an ideal readout of memory integration that minimizes potential on-the-fly retrieval strategies that might inflate rates of integration (*Abolghasem et al., 2023*). RT measures were queried as a function of cortical similarity and hippocampal–cortical connectivity.

In two experiments, participants viewed sequences of three objects in an incidental encoding task. Each sequence was constructed such that the first two objects always appeared consecutively (A and B), and the third object alternated between two objects ($C_1$ and $C_2$; *Figure 1C*). We reasoned that after repeated exposures to these sequences, participants would come to anticipate the presentation of two different objects ($C_1$ and $C_2$) upon encountering the beginning of each sequence (A and B). This expectation would give rise to a link between objects $C_1$ and $C_2$ strictly based on their shared antecedents. Thus, although $C_1$ and $C_2$ never co-occurred, we predicted that the two objects would become associated through their overlapping preceding A and B objects. We tested this prediction with a novel recognition priming task, which was developed to assess the implicit behavioral integration of overlapping sequences while minimizing intentional retrieval strategies at test. While we considered the priming task to be the main, planned index of behavioral integration, we also included an exploratory explicit memory test for comparison with the implicit test. Including both implicit and explicit memory tests also enabled us to explore changes in memory behaviors that may be underpinned by different memory processes (*Henke, 2010*; *Abolghasem et al., 2023*).

As a preview, we observed a delay-dependent priming effect such that behavioral integration of C objects emerged 24 hr after learning. We developed Experiment 2 to replicate this effect and investigate its relationship with neural signatures of systems consolidation. In this experiment, the same behavioral tasks were completed while participants underwent fMRI (note that the explicit test was modified; see Results for more details). We investigated how the observed pattern of behavioral integration related to pattern similarity in mPFC as well as changes in rest connectivity between posterior hippocampus and LOC. We predicted that neural patterns in mPFC would become more correlated for objects from overlapping sequences, but only after a period of consolidation. We further predicted that correlations between the hippocampus and LOC in rest connectivity would increase after learning, and that both measures would relate to the extent of behavioral priming participants exhibited after a delay. Finally, we examined the relationship between mPFC similarity and rest connectivity and their association with behavioral priming.

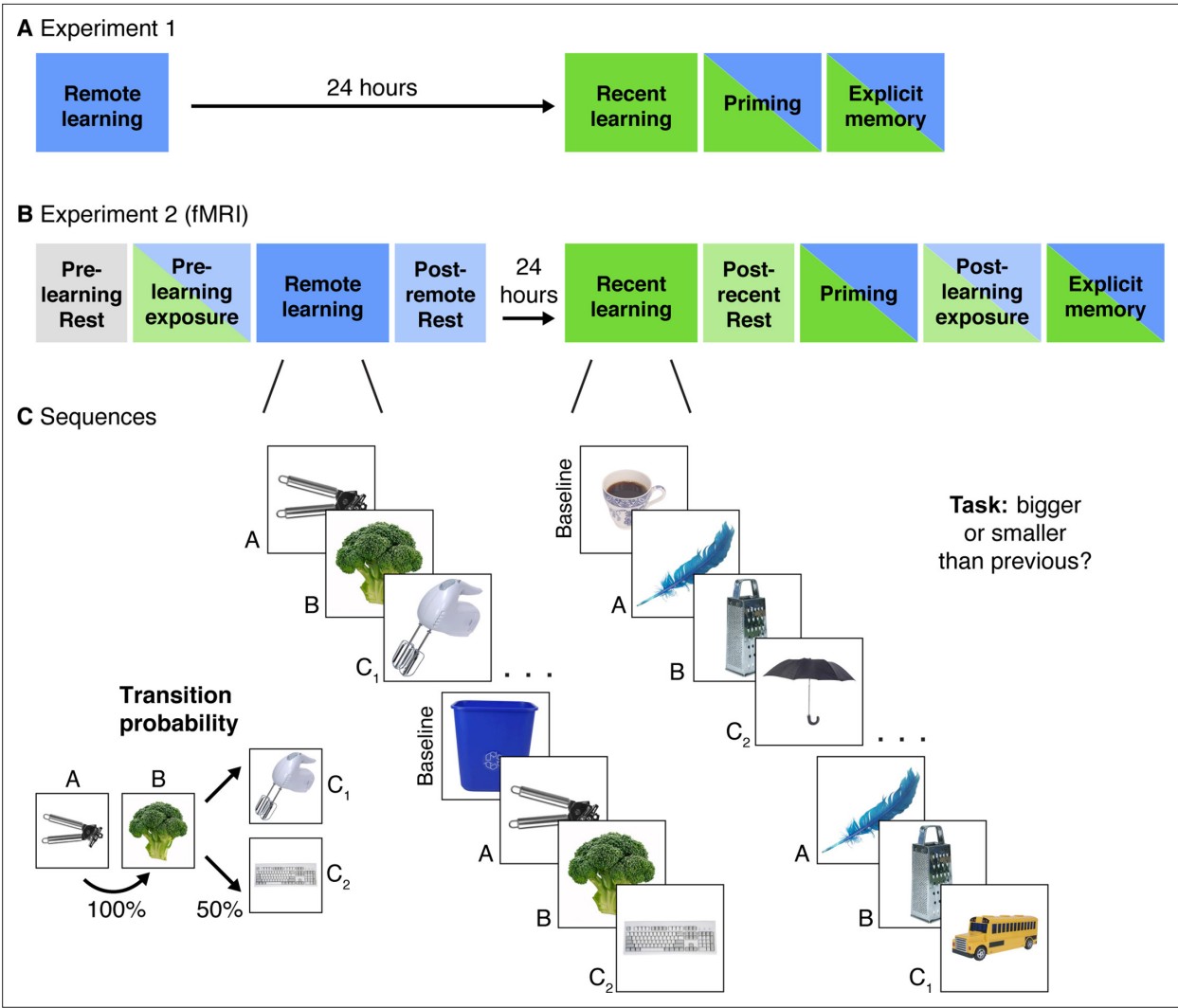

**Figure 1.** Experimental Design. (**A**) Experiment 1 design. Participants completed two learning sessions separated by 24 hr (blue = remote, green = recent learning). (**B**) Experiment 2 design. The timing, instructions, and procedures for the learning and priming tasks were identical to Experiment 1, with a modified but conceptually similar explicit memory task. Participants also completed pre- and post-learning exposure scans where all objects from both learning sessions were presented in a randomly intermixed order. Three rest scans were added: a pre-learning scan at the start of Day 1, a post-remote learning scan on Day 1, and a post-recent learning scan on Day 2. (**C**) In each learning session, participants performed a cover task as they viewed objects embedded in triplets. All triplets were composed of two images that always appeared back-to-back (A and B) and two images that alternated following the AB pair ($C_1$ and $C_2$) with equal probability. Baseline objects were randomly inserted between triplets. Separate image sets were presented in each learning session. After the recent learning session, participants completed recognition priming and explicit memory tests over all objects intermixed from the two learning sessions (see *Figure 2* for details of each test).

The online version of this article includes the following source data and figure supplement(s) for figure 1:

**Figure supplement 1.** AB recognition priming and explicit integration.

**Figure supplement 1—source data 1.** Response times and accuracy during learning.

**Figure supplement 2.** Average BOLD signal for predictable and unpredictable items across the eight learning runs, averaged over the recent and remote learning sessions.

**Figure supplement 2—source data 1.** Average BOLD signal during learning.

## Results
### Learning

During learning, participants were instructed to evaluate if the currently viewed object was bigger or smaller than the previous one. We used participants' RTs to assess learning of the statistical structure

embedded in the learning task and to confirm that sequence learning did not differ across the recent and remote learning sessions. Because B predictably followed A and either $C_1$ or $C_2$ predictably followed B, we reasoned that responses to B and C objects would be facilitated after repeated exposures to their sequential structure, resulting in decreased RTs relative to A and baseline objects, which were unpredictable. As an index of learning, we computed the RTs of participant's size judgments as a function of whether the response of each object could be predicted by the presentation of the prior one.

## Experiment 1

We computed the median RT of all A and baseline objects (unpredictable) and the median RT of all B and C objects (predictable), for each participant's 16 runs during each learning session. All trials were included regardless of accuracy for the size judgments (see Methods for more information). These values were entered into a mixed-effects linear model with repetition (continuous: 1–16), learning session (discrete: recent, remote), predictability (discrete: predicted, unpredicted), and their interactions as independent variables. This model revealed an effect of repetition ($F_{(1, 4448)}$ = 1070.17, p < 0.001), reflecting faster responses times as learning progressed, and an effect of predictability ($F_{(1, 232.4)}$ = 4.48, p = 0.04) such that participants responded more quickly to predictable objects over unpredictable ones. These two effects were qualified by an interaction ($F_{(1, 4448)}$ = 104.73, p < 0.001), reflecting a steeper drop in RTs over learning for predictable objects relative to unpredictable ones (***Figure 1— figure supplement 1A***). This confirms that participants learned the structure of the sequences, as by the end of learning, responses were more facilitated by predictable objects over unpredictable ones.

The same model also revealed differences between the recent and remote learning sessions, with an effect of learning session ($F_{(1, 119.9)}$ = 17.63, p < 0.001) and an interaction between repetition and learning session ($F_{(1, 4448)}$ = 4.57, p = 0.03). These effects were driven by overall slower RTs and a steeper change in RTs across learning in the remote session relative to the recent one, suggesting that responses in the recent session were in part driven by acclimation to the task. Critically, there was no interaction between learning session and predictability or between learning session, predictability, and repetition (both F's < 2.10, both p's > 0.14) suggesting that participants became sensitive to the sequence structure and were faster to respond to predictable objects over unpredictable ones during both sessions.

Average accuracy on the size judgment task was high (92.6%; SD = 12.5%). Changes in accuracy over learning were analyzed with the same mixed-effects linear model with repetition (continuous: 1–16), learning session (discrete: recent, remote), predictability (discrete: predicted, unpredicted), and their interactions as independent variables (***Figure 1—figure supplement 1B***). On average, accuracy did not reliably differ by learning session, repetition, or predictability (all F's < 2.57, all p's > 0.11). However, this model revealed several interactions: between repetition and predictability ($F_{(1, 4448)}$ = 7.82, p = 0.005), with higher accuracy for predictable objects over unpredictable objects toward the end of learning relative to the beginning; between learning session and predictability ($F_{(1, 4448)}$ = 4.07, p = 0.04), with a more pronounced difference in accuracy by predictability in the recent session over the remote; and between repetition and learning session ($F_{(1, 4448)}$ = 9.72, p = 0.002), with a more pronounced increase in accuracy over learning in the remote session relative to the recent one. Taken together, this suggests that although size comparisons were overall very accurate, responses were influenced by the predictability of the objects and the order of the learning sessions.

## Experiment 2

Participants exhibited the same changes in RTs as in Experiment 1 (***Figure 1—figure supplement 1A***). Participants became faster across repetitions ($F_{(1, 1459)}$ = 354.21, p < 0.001) and for predicted relative to unpredictable objects ($F_{(1, 43.06)}$ = 19.00, p < 0.001). Critically, the interaction between predictability and repetition was replicated ($F_{(1, 1459)}$ = 34.66, p < 0.001) with more facilitated responses across repetitions for predictable relative to unpredictable objects. The same effects of learning session were present as well: faster RTs during recent relative to remote sessions ($F_{(1, 40.69)}$ = 22.66, p < 0.001), and a larger facilitation of RTs across repetitions in the remote session over the recent session ($F_{(1, 1459)}$ = 32.16, p < 0.001). As in Experiment 1, there were no other interactions with learning session (both F's < 2.65, both p's > 0.10), suggesting that learning was equivalent across the two sessions.

Average accuracy on the size judgment task was consistently high (95.3%; SD = 7.4%) and changes in accuracy over learning were largely similar to those observed in Experiment 1 (*Figure 1—figure supplement 1B*). Specifically, accuracy was modulated by an interaction between repetition and predictability ($F_{(1, 1459)}$ = 8.61, p = 0.003), reflecting higher accuracy for predictable over unpredictable objects toward the end of learning, and an interaction between repetition and learning session ($F_{(1, 1459)}$ = 5.66, p = 0.02), with a more pronounced increase in accuracy over learning in the remote session relative to the recent one. There were no reliable effects of learning session or predictability and no other interactions (all $F$'s < 0.99, all p's > 32), with the exception of an effect of repetition ($F_{(1, 1459)}$ = 713, p = 0.008) not observed in Experiment 1.

## Recognition

As a reminder, although $C_1$ and $C_2$ were never experienced together in time, we predicted that the two objects would become associated through their overlapping preceding A and B objects, and that this across-sequence association would become strengthened over time. To test this prediction, we developed a novel recognition priming task in which participants viewed all objects from both learning sessions intermixed with novel foils and endorsed each as 'old' or 'new'. Unbeknownst to the participants, the order of the objects was manipulated to use response priming as a behavioral index of participant's implicit integration of C objects from overlapping sequences (*Figure 2A*, top). Specifically, participants viewed a baseline object, then a $C_1$ object, then the $C_2$ object from its overlapping sequence. Notably, during learning, these three objects were presented the same number of times and never appeared in sequence. We developed mixed-effects models to examine differences in RTs between $C_2$ objects, which were preceded by and thus primed by $C_1$, and RTs for $C_1$ objects, which were preceded by baseline objects and thus served as a control comparison (see Methods). To be included in this priming analysis, each pair of C objects and the preceding baseline object must have been correctly endorsed as 'old', meaning their responses during recognition were matched and thus could not be slowed by pressing a different button. Furthermore, each object only appeared once during recognition, as repeated presentations of objects may facilitate RTs and mask potential priming effects. Note that $C_1$ and $C_2$ are not differentiable to participants, as during learning, they both follow B an equal number of times and are presented within their sequence in a randomized and intermixed order. We assigned them with separate labels for the sole purpose of clarifying the conditions of the priming manipulation.

### Experiment 1

Recognition accuracy across all objects approached perfect performance, as measured by A (*Zhang and Mueller, 2005*), a non-parametric measure of sensitivity that integrates hits and false alarms and can accommodate perfect performance (remote: mean A = 0.967, SD = 0.043; recent: mean A = 0.967, SD = 0.045). A Wilcox signed-ranks test indicated no detectable difference across learning sessions (V = 683.5, p = 0.48).

Because each participant had different numbers of trials included, mixed-effects linear models were computed with RT (continuous) as the dependent variable and order (discrete: primed, control) as a predictor, separately for recently and remotely learned objects (*Figure 2A*, bottom left). This decision reflected our a priori prediction that remotely learned sequences would be more strongly integrated over recently learned ones. Indeed, we found faster RTs for primed objects relative to unprimed objects for the remotely learned objects ($F_{(1, 674.43)}$ = 7.97, p = 0.005) but not the recently learned objects ($F_{(1, 69.12)}$ = 0.37, p = 0.54). This finding suggests that objects that are linked by overlapping preceding sequences become associated over time. To assess if these effects were different across the two learning sessions, a mixed-effects model was computed with order (discrete: primed, control), learning session (discrete: recent, remote), and their interaction as predictors. This revealed an effect of order ($F_{(1, 70.07)}$ = 5.40, p = 0.02), no reliable effect of session ($F_{(1, 60.45)}$ = 3.40, p = 0.07), and no reliable interaction ($F_{(1, 1295)}$ = 2.54, p = 0.11). While we observed reliable priming only after 24 hr, the lack of a statistical interaction suggests caution in interpreting this as an effect that emerges solely after a delay.

In this task, the order of trials was the same for each sequence: a baseline object, $C_1$ (control condition), and then $C_2$ (primed condition). The advantage of this approach is that every sequence was tested without repeating any images, as repetition could facilitate responses and mask potential

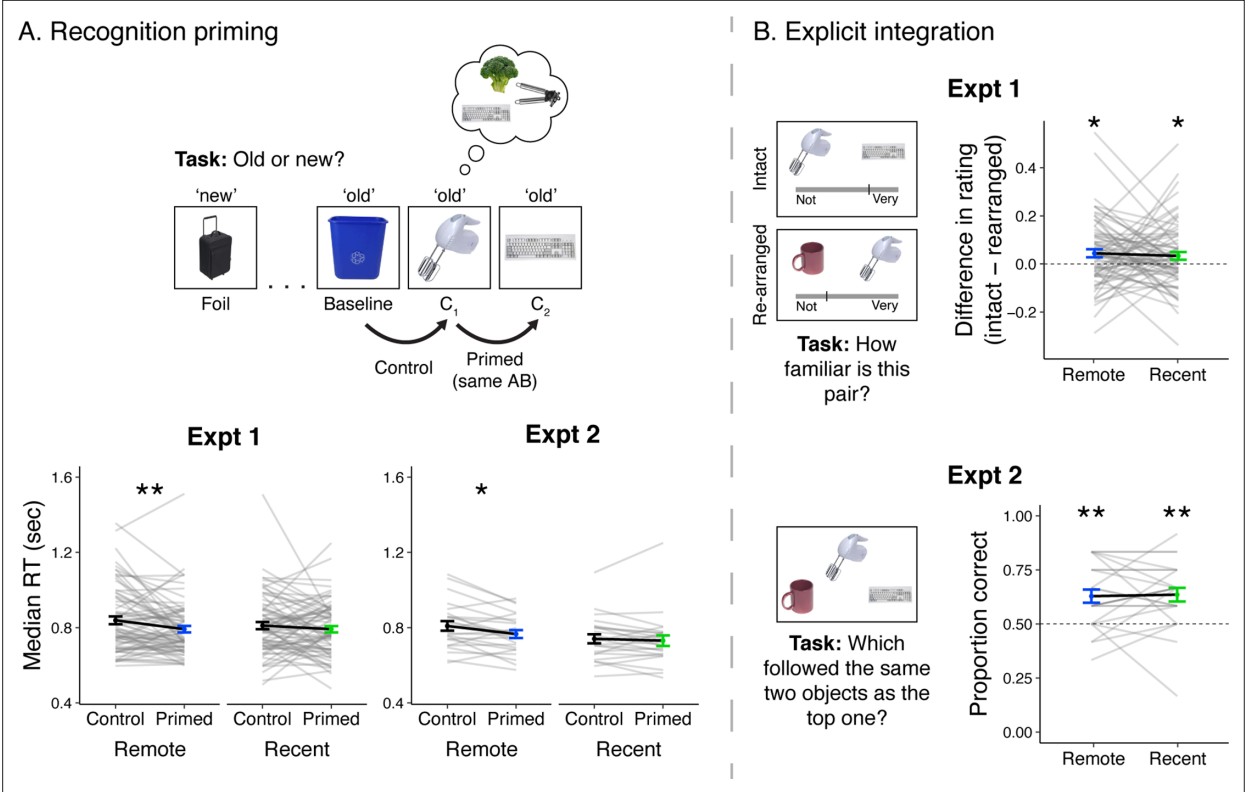

**Figure 2.** Recognition priming and explicit integration. (**A**, top) Recognition priming design. Studied and new objects were presented pseudo-randomly such that each $C_2$ object followed the $C_1$ object from the same sequence, and the $C_1$ object followed a baseline object from the same learning session. The behavioral integration of $C_1$ and $C_2$ was operationalized as the decrement in response time to $C_2$ (preceded by $C_1$) relative to $C_1$ (preceded by a baseline object). (A, bottom) Recognition priming results. Statistics reflect results from trial-level mixed-effects models, but participants' responses are aggregated to facilitate visualization of the effects. Black lines represent group averages. Gray lines represent median response times for each participant. Error bars indicate standard error of the mean (SEM) across participants. ** indicates p < 0.01; * indicates p < 0.05. (**B**, left) Explicit integration design. In Experiment 1, participants viewed intact and re-arranged pairs of C objects and rated their familiarity on a continuous scale from 'Not familiar at all' to 'Very familiar'. All foils for re-arranged pairs were C objects from a different sequence learned in the same session. The average difference in rating for intact versus re-arranged pairs served as a measure of explicit integration across overlapping sequences. In Experiment 2, participants performed a 2-alternative forced choice (2AFC) task with a C object as the cue and were asked to choose which object followed the same pair of two objects as the cue. Foil images were C objects from a different sequence learned in the same session. (**B**, right) Explicit integration results. Black lines represent group averages. Gray lines represent participants. Error bars indicate standard error of the mean (SEM) across participants. Statistics reflect one-sample *t*-tests against chance performance. ** indicates p < 0.01; * indicates p < 0.05. Plots can be reproduced with data in '*Figure 2—source data 1*' and '*Figure 2—source data 2*'.

The online version of this article includes the following source data and figure supplement(s) for figure 2:

**Source data 1.** Recognition priming.

**Source data 2.** Explicit integration.

**Figure supplement 1.** Plots can be reproduced with '*Figure 2—figure supplement 1—source data 1*' and '*Figure 2—figure supplement 1—source data 2*'.

**Figure supplement 1—source data 1.** AB recognition priming.

**Figure supplement 1—source data 2.** AB explicit integration.

priming effects. However, if participants improved at this task with practice, faster responses over the course of the task could explain the above priming effects—although notably, no priming was observed for recently learned sequences, despite being susceptible to the same practice effect. Regardless, we next ruled out the possibility that practice effects contributed to the priming effect for remotely learned sequences by conducting the same mixed-effects model as above with trial order as an additional continuous predictor. This analysis revealed an effect of trial order such that responses grew faster over time ($F_{(1, 710.22)} = 9.74$, $p = 0.002$). Critically, when accounting for this effect, we still observed faster RTs for primed objects relative to unprimed objects ($F_{(1, 190.11)} = 6.732$, $p = 0.01$).

## Experiment 2

Similar to Experiment 1, recognition accuracy across all objects was consistently high (remote: mean $A$ = 0.977, SD = 0.018; recent: mean $A$ = 0.983, SD = 0.013) with a small but reliable decrease in accuracy for remotely learned objects relative to recently learned ones ($V$ = 12.0, p = 0.04).

Priming analyses for participants in Experiment 2 primarily replicated findings from Experiment 1 (*Figure 2A*, bottom right). As in Experiment 1, we found faster RTs for primed trials relative to control trials, for objects learned in the remote session ($F_{(1, 224.18)}$ = 5.70, p = 0.02) but not the recent session ($F_{(1, 244.16)}$ = 0.54, p = 0.46). However, a trial-level mixed-effects model including trials from both sessions revealed an effect of order ($F_{(1, 468.92)}$ = 4.65, p = 0.03), an effect of learning session ($F_{(1, 23.63)}$ = 5.81, p = 0.02), and no reliable interaction ($F_{(1, 469.00)}$ = 1.27, p = 0.26). When accounting for trial order in the remote condition, there was no reliable change in RTs across the task ($F_{(1, 228.07)}$ = 1.03, p = 0.31), and the effect of faster RTs for primed trials remained significant ($F_{(1, 88.91)}$ = 5.26, p = 0.02).

Taken together, RTs from both cohorts reflect a behavioral association of objects from overlapping sequences, despite never having been experienced at the same moments in time. This association only emerges for the remotely learned sequences.

## Explicit integration

Our primary, planned predictions involved the recognition priming task, as an implicit measure of association would minimize any contributions of active strategies or control processes that may contribute to integration at retrieval. However, we also included exploratory explicit tests of memory integration for comparison to other studies that report such tests (*Figure 2B*, left). In Experiment 1, we explored a novel task that provided us with a continuous measure of integration. In Experiment 2, we modified this task to a 2-alternative forced choice (2AFC) format more commonly used to investigate neural measures underlying integration (*Preston et al., 2004*; *Zeithamova et al., 2012*), to enable more direct comparison to prior work.

### Experiment 1

In this task, participants viewed pairs of C objects that either followed the same A and B objects (Intact) or were paired with different A and B objects (Re-arranged) during learning. They reported the strength of their associative recognition of these pairs on a sliding scale. Behavioral integration was operationalized as the average difference in familiarity between Intact and Re-arranged pairs. A value of 0 indicates no discrimination and 1 indicates maximal discrimination between the two conditions. We found significant evidence for integration (remote: mean = 0.044, SD = 0.142, $t_{(70)}$ = 2.63, p = 0.01; recent: mean = 0.034, SD = 0.139, $t_{(70)}$ = 2.04, p = 0.046) with no reliable difference across learning sessions ($t_{(70)}$ = 0.46, p = 0.65; *Figure 2B*, top right).

### Experiment 2

Here, we tested explicit integration with a 2AFC task with a C object as the cue, the C object from the overlapping A and B sequences as the target, and a C object from a different sequence as the foil. We computed the proportion of trials in which participants chose the overlapping C object, with performance at 0.5 indicating chance-level performance. Integration was above chance for objects learned remotely (mean = 0.628, SD = 0.149, $t_{(23)}$ = 4.21, p < 0.001) and recently (mean = 0.635, SD = 0.155, $t_{(23)}$ = 4.28, p < 0.001) with no difference across the two learning sessions ($t_{(23)}$ = −0.22, p = 0.83; *Figure 2B*, bottom right). These results serve as a conceptual replication of Experiment 1, as both cohorts exhibited weak integration between C objects that did not change over time. Interestingly, these results differ from the priming findings, which revealed behavioral integration only after a delay. Potential accounts for this discrepancy are considered in the Discussion.

## Experiment 2: pattern similarity

Participants in Experiment 2 completed a modified set of procedures from Experiment 1 while undergoing fMRI (*Figure 1B*, see Methods). Specifically, their procedure included a pre- and post-learning exposure phase in which they viewed all object images from the recent and remote learning sessions, and three scans capturing periods of rest occurring pre-learning, post-recent learning, and post-remote learning. In this section, we focus on the pre- and post-learning exposure scans (*Figure 3A*) in which participants pressed a button when a hash tag appeared in any image. Presenting images before and

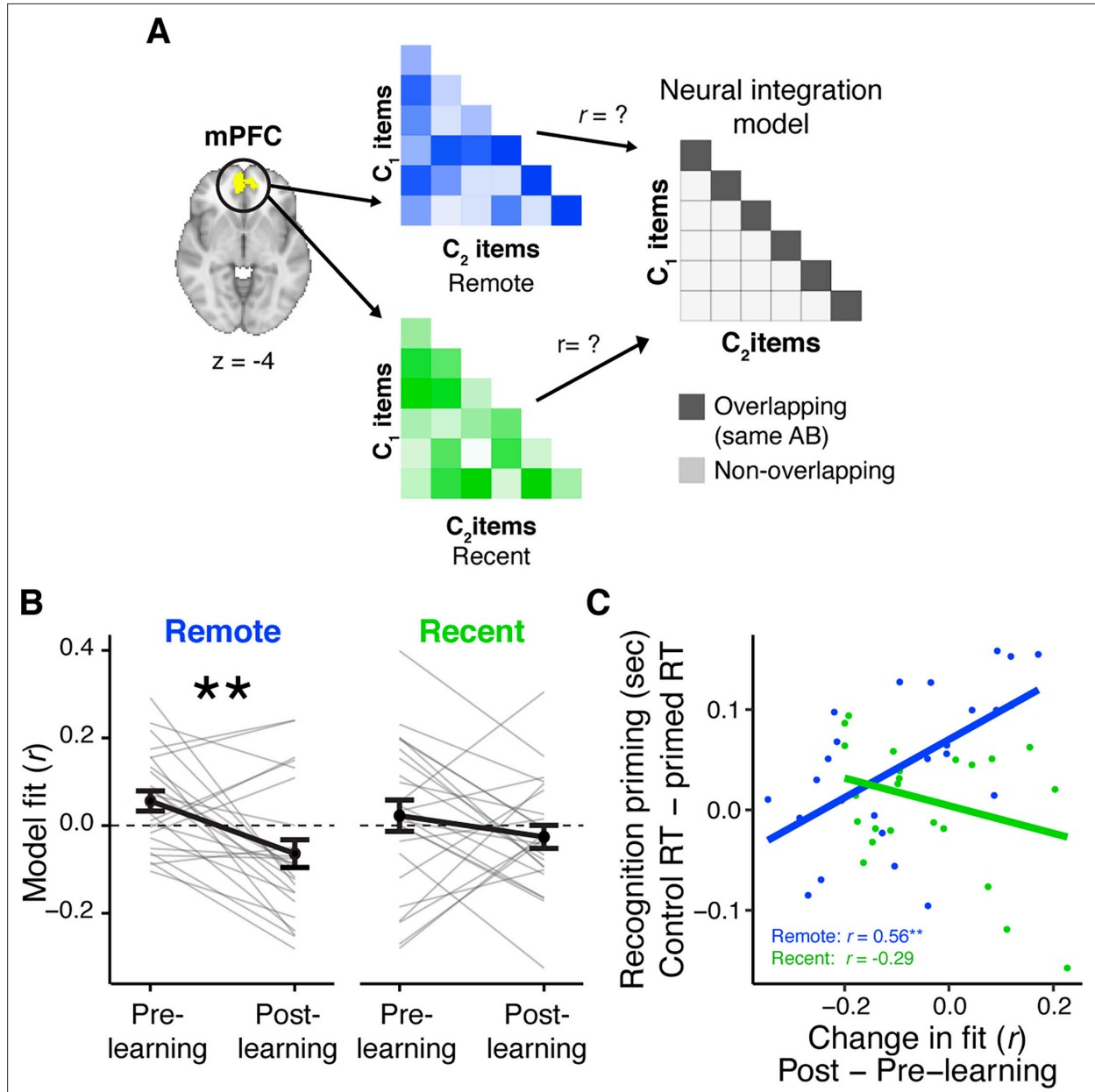

**Figure 3.** Learning- and consolidation-dependent changes in mPFC neural integration. (**A**) Analysis approach for exposure scans. Vectors of multi-variate activation were extracted from a medial prefrontal cortex (mPFC) region of interest (ROI) for each C object and correlated with vectors for all C objects from overlapping and distinct sequences viewed in the same learning session (recent and remote). These correlation matrices were correlated with an 'integration model' reflecting greater similarity for C objects belonging to an overlapping sequence (i.e. objects that followed the same AB sequence). This analysis was repeated for both exposure phases and subtracted (post- minus pre-learning exposure) to create change scores. (**B**) Average fit to the integration model for the pre- and post-learning exposure phases in mPFC, separately for recently and remotely learned sequences. Values <0 indicate a worse fit to the model, meaning more differentiation between C objects from overlapping sequences relative to C objects from different sequences (see also *Figure 3—figure supplement 1*). Gray lines indicate participants. Black line indicates group average. Error bars reflect standard error of the mean (SEM). ** indicates p < 0.01. (**C**) Correlation between the average change in the fit to the integration model in mPFC and average recognition priming across participants, separately for recent and remote learning. Dots indicate participants. Lines indicate line of best fit. Plots can be reproduced with '*Figure 3—source data 1*.csv' and '*Figure 3—source data 2*.csv'.

The online version of this article includes the following source data and figure supplement(s) for figure 3:

**Source data 1.** mPFC fit to neural integration model.

**Source data 2.** mPFC neural integration related to recognition priming.

*Figure 3 continued on next page*

*Figure 3 continued*

**Figure supplement 1.** Average similarity across $C_1$ and $C_2$ items in overlapping sequences (left) and non-overlapping sequences (right), separately for the pre- and post-learning exposure phases and for recent and remotely learned sequences.

**Figure supplement 1—source data 1.** mPFC similarity for overlapping and non-overlapping sequences.

**Figure supplement 2.** mPFC similarity related to recognition priming.

**Figure supplement 2—source data 1.** mPFC similarity related to recognition priming.

**Figure supplement 3.** Neural integration in LOC.

**Figure supplement 3—source data 1.** LOC fit to neural integration model.

**Figure supplement 3—source data 2.** LOC neural integration related to recognition priming.

**Figure supplement 3—source data 3.** LOC similarity for overlapping and non-overlapping sequences.

**Figure supplement 4.** Neural integration of A and B items.

**Figure supplement 4—source data 1.** Neural integration of A and B in anterior hippocampus.

**Figure supplement 4—source data 2.** Neural similarity for overlapping and non-overlapping A and B sequences in anterior hippocampus.

**Figure supplement 4—source data 3.** Neural integration of A and B in LOC.

**Figure supplement 4—source data 4.** Neural similarity for overlapping and non-overlapping A and B sequences in LOC.

after learning enables us to extract 'snapshots' of the pattern of activity evoked by each image before and after participants learned their temporal associations. Specifically, we were interested in quantifying changes in the similarity in patterns of activity evoked by C objects as a function of whether they were learned in overlapping or distinct sequences (e.g. followed by the same or different AB images) and when they were learned (either in the recent or remote learning sessions). We also examined their relationship with the recognition priming effects observed in those participants. The pseudo-random trial order allowed us to statistically separate responses to individual images (*Schapiro et al., 2012*), and $C_1$ and $C_2$ objects were presented in separate exposure scans to ensure that their similarity was not driven by temporal proximity (see Methods for more details).

## Learning-related and time-dependent changes in similarity

We quantified changes in similarity between all C objects by extracting patterns of activation for each C object and compiling the correlation between all patterns into a matrix, separately for sequences learned recently or remotely and separately for the pre- and post-exposure phases. We correlated these four matrices with a model matrix postulating greater similarity between C objects that followed the same AB sequence relative to those that followed different AB objects ('neural integration model'; *Figure 3A*). We then compared the pre-learning fit from the post-learning fit for recently and remotely learned objects, where a significant increase in fit would indicate that C objects following the same AB sequence became more similar to each other relative to C objects from other sequences. In contrast, a decrease in fit would indicate that C objects became more differentiated. As a reminder, the pre-learning exposure phase took place immediately before remote learning on Day 1, and the post-learning exposure phase took place after recent learning on Day 2. This design allows us to dissociate changes in similarity that emerge from sequence learning alone (recent learning), versus from learning followed by a period of consolidation (remote learning).

In this planned analysis, we predicted greater similarity between C objects in mPFC but only for remotely learned sequences. To this end, we applied this analysis to multi-voxel patterns in a functionally defined mPFC region from the learning scans (see Methods). This region and others exhibited greater BOLD signal for predictable objects (e.g. B and C objects) relative to unpredictable objects (i.e. A and baseline objects), when considering all learning scans in both sessions (*Figure 1—figure supplement 2A*). We chose this contrast to functionally define regions of interest (ROIs) because we reasoned that cortical regions that were sensitive to the temporal regularities generated by the sequences may be further involved in their offline consolidation and long-term storage (*Danker and Anderson, 2010*; *Davachi and Danker, 2013*; *McClelland et al., 1995*). Surprisingly, in mPFC, the change in fit between the correlation matrix reflecting neural similarity and the model was significantly negative for the remotely learned objects ($t_{(23)} = -3.06$, p = 0.006; Bonferroni-corrected p = 0.024; *Figure 3B*). This decrease remained when accounting for variation in BOLD signal (see Methods for more details). A negative change score reflects a *decrease* in fit between the multi-voxel patterns

and the integration model in the post-learning relative to pre-learning snapshots, which means that over a combined period of learning and consolidation, voxels in mPFC *differentiated* objects that appeared in overlapping sequences, contrary to our prediction. We confirmed this by computing the average similarity separately for C objects in overlapping and non-overlapping sequences, and separately for pre- and post-learning snapshots. We found a reliable decrease in similarity that was selective to objects from overlapping sequences, and no reliable change in similarity among objects in non-overlapping sequences (*Figure 3—figure supplement 1*). There was no reliable change in fit to the neural integration model for recently learned objects ($t_{(23)} = -1.37$, $p = 0.18$), however there was no reliable difference in the extent of the change in fit between recent and remote learning ($t_{(23)} = -1.42$, $p = 0.17$).

## Relationship between mPFC similarity and priming

Next, we asked whether changes in mPFC pattern similarity tracked the recognition priming observed for objects whose sequences were learned 24 hr prior. To do this, we quantified the priming effect for each participant as the median of the difference in RTs between responses to primed and unprimed C objects, where a greater difference score indicates a stronger priming effect. We then correlated this score with participants' change in fit to the neural integration model from the pre- to post-learning exposure phases. We computed this correlation separately for the objects learned recently and remotely (*Figure 3C*). We found that participants with *increased similarity* among C pairs from overlapping sequences exhibited a stronger priming effect, but only for pairs learned remotely ($r_{(22)} = 0.57$, $p = 0.004$, Bonferroni-corrected $p = 0.016$). This relationship was not detectable for pairs learned recently ($r_{(22)} = -0.30$, $p = 0.15$) and the difference in the two correlations was significant ($z = 3.07$, $p < 0.001$). Furthermore, the correlation between change in mPFC similarity and priming was driven specifically by a change in similarity in the overlapping sequences learned in the remote session (*Figure 3—figure supplement 2*). This suggests that across participants, the extent of neural integration in mPFC reflects the extent of behavioral integration across overlapping sequences, but this association only emerges after a period involving consolidation.

## Control analyses

Here, we report steps taken to ensure that the observed changes in pattern similarity were not influenced by variability in BOLD signal across different patterns. First, our pattern similarity approach relied on point-biserial correlation, a special case of Pearson correlation that is invariant to average levels of BOLD signal across patterns. Thus, *z*-scoring across all voxels within a pattern would give rise to the exact correlation matrices as reported in the un-transformed main findings. Second, we *z*-scored BOLD signal of each voxel across all patterns, which aims to mitigate the influence of inordinately noisy voxels by down-weighting their extreme values across all trials (*Kuhl and Chun, 2014*; *Richter et al., 2016*). This did not meaningfully change the reported results. This suggests that our findings are not influenced by variation in average BOLD signal.

We also developed an analysis to account for cases where $C_1$ and $C_2$ objects exhibited differences in average BOLD signal, reasoning that equivalent levels of BOLD signal (both high or both low) may give rise to higher similarity between two objects. To do this, we calculated the absolute value of the difference in average BOLD signal between each $C_1$ and $C_2$ object. We then re-analyzed the change in fit between mPFC similarity and the neural integration model, and the extent that this change correlated with recognition priming, while accounting for this new measure using partial correlations (see Methods for details of this analysis). When accounting for the difference in BOLD signal, we found that the correlation between neural similarity and the integration model was significantly negative for the remotely learned objects ($t_{(23)} = -3.08$, $p = 0.005$) and not for recently learned objects ($t_{(23)} = -1.37$, $p = 0.18$), with no reliable difference between the two ($t_{(23)} = -1.51$, $p = 0.17$). Furthermore, participants with increased similarity among C pairs exhibited a stronger priming effect, but only for pairs learned remotely ($r_{(22)} = 0.58$, $p = 0.003$) and not recently ($r_{(22)} = -0.31$, $p = 0.15$). In summary, we replicated the observed findings: (1) mPFC similarity decreased after learning, particularly after remote learning, and (2) participants with a greater change in similarity in mPFC after learning exhibited stronger recognition priming, only for the remotely learned objects. This suggests that the reported findings are not driven by average BOLD signal.

## Changes in similarity in other regions

In addition to our planned analysis of mPFC, we conducted exploratory analyses of learning- and consolidation-related changes in similarity in other regions. Prior work has observed neural integration in category-selective cortical regions immediately after learning (*Richter et al., 2016*; *Tompary and Davachi, 2017*; *Wing et al., 2020*). Using the same neural integration model as with mPFC, we analyzed patterns of activity in LOC, a region sensitive to object stimuli. We found that C objects belonging to overlapping sequences grew more similar to each other after recent learning ($t_{(23)}$ = 3.10, p = 0.005, Bonferroni-corrected p = 0.02) but not for the remotely learned objects ($t_{(23)}$ = 0.38, p = 0.71), with no reliable difference across the two ($t_{(23)}$ = 1.98, p = 0.06, *Figure 3—figure supplement 3A*). Interestingly, the average similarity between objects from all sequences increased in LOC, but the improved model fit for recently learned sequences was driven by a larger increase in similarity for objects from overlapping sequences relative to non-overlapping ones (*Figure 3—figure supplement 3C*). Unlike in mPFC, the extent of this change did not relate to behavioral priming across participants (both *r*'s < 0.3, both p's > 0.14; *Figure 3—figure supplement 3B*). Neither anterior nor posterior hippocampal ROIs exhibited changes in their fit to the neural integration model (all *t*'s < 0.94, all p's > 0.35).

Taken together, we find that although on average, objects that appeared separately in time, but shared overlapping antecedents, are integrated in LOC immediately after learning but are differentiated in mPFC after 24 hr. However, despite this differentiation at the group-level, participants with stronger neural integration in mPFC exhibited facilitated behavioral integration as reflected by stronger recognition priming.

## AB integration and similarity

While the primary focus of this experiment was investigating behavioral and neural integration *across* sequences with overlapping regularities, we also included tests of memory for AB pairs, to confirm that participants were able to learn those components of each sequence. We observed consistently high explicit memory for the pairs across both experiments, and implicit integration after a delay in Experiment 1 only (*Figure 2—figure supplement 1*). We also designed Experiment 2 to replicate findings of integration *within* sequences by assessing the similarity of A and B objects from overlapping versus different sequences (*Figure 3—figure supplement 4A*). For this analysis, we had one planned prediction, which was that AB pairs from the same sequences in the recent session would be more neurally integrated in the anterior hippocampus, as this would be a conceptual replication of prior work showing neural integration of temporally co-occurring shapes immediately after learning (*Schapiro et al., 2012*). Consistent with this prediction, we found greater similarity for AB pairs from the same sequences in recent learning in the anterior aspect of the hippocampus, but not the posterior aspect (*Figure 3—figure supplement 4B, C*). In an exploratory analysis, we also observed a similar pattern of effects in in lateral occipital cortex; however, in this region, all AB pairs become more similar to each other after learning, but this change was greatest for the overlapping AB pairs in the recent condition (*Figure 3—figure supplement 4D, E*).

## Experiment 2: resting-state connectivity

Having established a behavioral measure representing time-dependent integration across events and linking that behavior to a neural measure of integration, we next were interested in understanding whether these emerging signals were supported by active consolidation processes. As mentioned in the introduction, our planned analyses focused on changes in resting-state connectivity between the posterior hippocampus and category-selective LOC. Rest connectivity is operationalized here as the Pearson correlation between the time courses of two ROIs (*Figure 4A*).

### Learning-related changes in rest connectivity

To first examine broad changes resulting from sequence learning, we collapsed across both recent and remote learning and found a reliable increase in rest connectivity between posterior hippocampus and LOC ($t_{(22)}$ = 2.26, p = 0.03). This increase was driven primarily by an increase in connectivity after remote learning (occurring on Day 1) ($t_{(22)}$ = 2.70, p = 0.01, Bonferroni-corrected p = 0.065) rather than after recent learning (occurring on Day 2) ($t_{(22)}$ = 1.11, p = 0.28), although the extent of these increases were not reliably different from each other ($t_{(22)}$ = 1.23, p = 0.23). Notably, Day 1 of the experiment

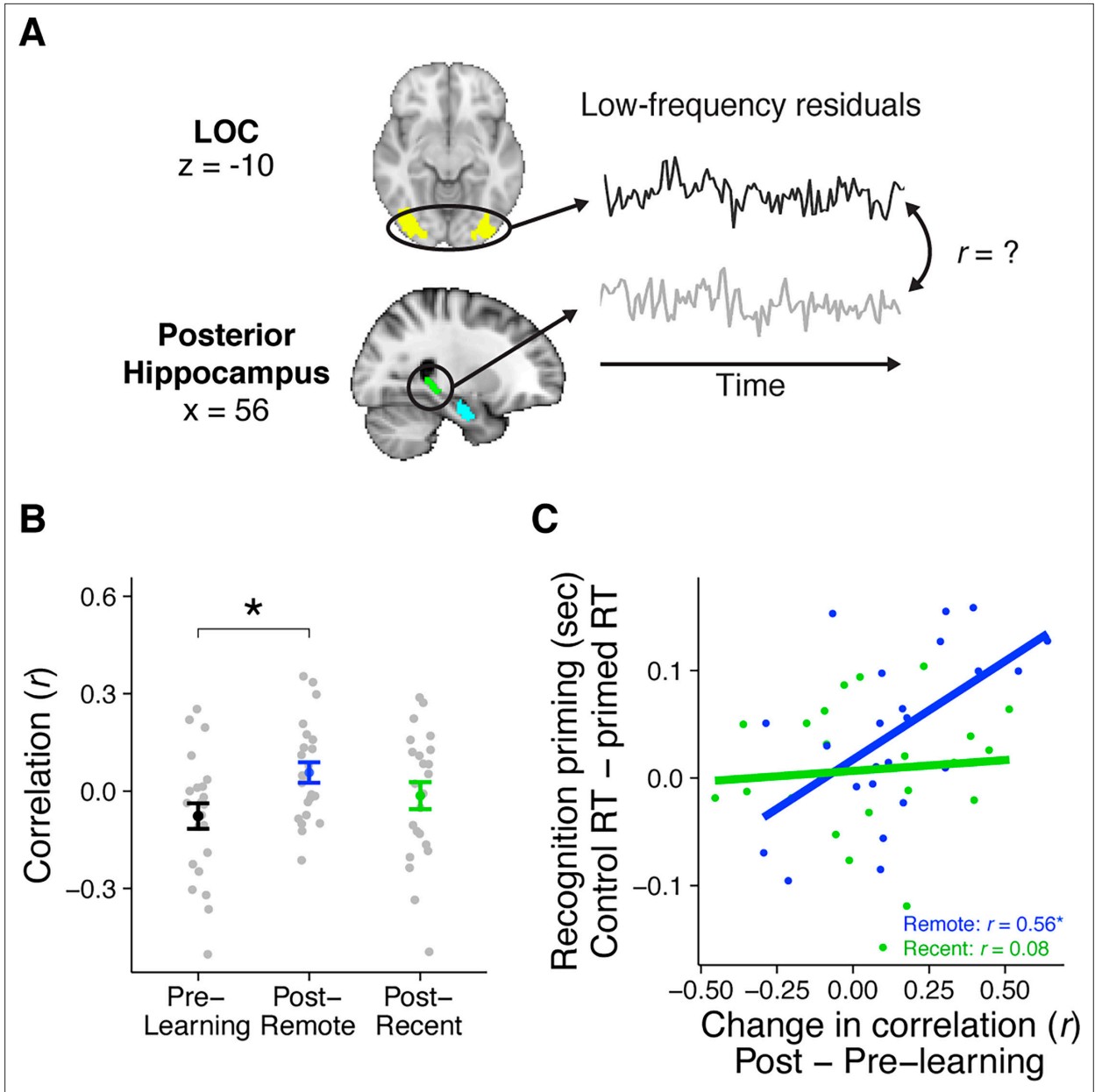

**Figure 4.** Rest connectivity between LOC and posterior hippocampus. (**A**) Analytic approach for rest scans. Rest scans were preprocessed, stripped of nuisance signals, and band-pass filtered. The mean residual signal was extracted from the posterior hippocampus and LOC for each volume of each scan. Rest connectivity was measured by correlating their mean time courses. (**B**) Average rest connectivity between LOC and posterior hippocampus. Gray dots indicate participants. Black, blue, and green dots indicate group averages. Error bars reflect standard error of the mean (SEM). * indicates p < 0.05. (**C**) Correlation between the change in LOC—posterior hippocampal rest connectivity and average recognition priming across participants, separately for recent and remote learning. Dots indicate participants. Lines indicate best fit. Plots can be reproduced with '*Figure 4—source data 1*.csv' and '*Figure 4—source data 2*.csv'.

The online version of this article includes the following source data and figure supplement(s) for figure 4:

**Source data 1.** Rest connectivity between LOC and posterior hippocampus.

**Source data 2.** Rest connectivity between LOC and posterior hippocampus related to recognition priming.

**Figure supplement 1.** Rest connectivity between LOC and posterior hippocampus related to recognition priming, separately for pre- and post-learning rest scans.

**Figure supplement 1—source data 1.** Rest connectivity between LOC and posterior hippocampus related to recognition priming, separately for pre- and post-learning rest scans.

**Figure supplement 2.** Rest connectivity between LOC and mPFC.

*Figure 4 continued on next page*

*Figure 4 continued*

**Figure supplement 2—source data 1.** Rest connectivity between LOC and mPFC.

**Figure supplement 2—source data 2.** Rest connectivity between LOC and mPFC, related to recognition priming.

(comprising pre-learning rest, remote learning, and post-remote learning rest) is most analogous to prior designs investigating learning-related changes in connectivity, in that all scans take place on the same day, in the same scan session, with no learning or experimental sessions. The observation that hippocampal–LOC connectivity increases reliably after remote learning is a conceptual replication of those findings (*Murty et al., 2017*; *Tambini et al., 2010*). This finding was selective to aspects of the posterior hippocampus, as there was no analogous change in rest connectivity between anterior hippocampus and LOC ($t_{(22)} = 0.23$, p = 0.82).

Since mPFC exhibited changes in neural patterns that reflect learning and consolidation of the sequences, we also explored whether rest connectivity with this region changed after learning. Surprisingly, there was an increase in rest connectivity between LOC and mPFC ($t_{(22)} = 2.25$, p = 0.03). This increase was driven primarily by an increase after recent learning, on Day 2, although this did not survive comparison for multiple corrections ($t_{(22)} = 2.31$, p = 0.03, Bonferroni-corrected p = 0.15). There was no corresponding change after remote learning ($t_{(22)} = 1.18$, p = 0.25), with no reliable difference across sessions ($t_{(22)} = -1.00$, p = 0.33; *Figure 4—figure supplement 2A*). There was no reliable change in rest connectivity between mPFC and either anterior or posterior hippocampus (both *t*'s < 1.23, both p's > 0.23), suggesting a selective role of sensory cortex in post-learning processing of the sequences consistent with prior work (*Murty et al., 2017*; *Tambini et al., 2010*).

## Relationship between rest connectivity and priming

As past work has observed that learning-related changes in rest connectivity are related to later memory for the preceding learned items (*Murty et al., 2017*; *Tambini et al., 2010*; *Tompary et al., 2015*), we next investigated whether changes in rest connectivity between posterior hippocampus and LOC related to our measures of integration as evidenced by recognition priming (*Figure 4C*). To do this, we conducted two across-participant correlations between the change in rest connectivity (post- minus pre-learning) and median recognition priming of sequences: one from the remote learning session, and another from the recent learning session. We found that participants with a greater change in posterior hippocampal–LOC connectivity exhibited larger priming effects only for sequences from the remote learning session ($r_{(21)} = 0.56$, p = 0.005, Bonferroni-corrected p = 0.025) and not the recent learning session ($r_{(21)} = 0.09$, p = 0.69). This relationship between connectivity and behavior was not significantly different across the two sessions ($z = 1.68$, p = 0.09), suggesting caution in interpreting the relationship between connectivity and priming as solely emerging after a delay.

We conducted several control analyses to investigate the specificity of the relationship between changes in rest connectivity and recognition priming from the remote session. First, rest connectivity from the recent session did not relate to remote priming, and rest connectivity from the remote session did not relate to recent priming (both *r*'s < 0.10, both p's > 0.65), suggesting that learning-related changes in rest were related solely to the memoranda learned in the intervening session. Second, LOC rest connectivity with neither mPFC (*Figure 4—figure supplement 2B*) nor anterior hippocampus related to priming from either session (all *r*'s < 0.23, all p's > 0.28). Third, rest connectivity between LOC and posterior hippocampus did not relate to any explicit memory measures for $C_1$ and $C_2$ pairs at either time point (both *r*'s < -0.32, both p's > 0.13). Together, these observations demonstrate that the long-term implicit integration across overlapping sequences is selectively related to immediate post-learning rest connectivity between the posterior hippocampus and LOC.

## Accounting for same- versus across-session change scores

A critical difference between the recent and remote conditions is that the pre-learning rest scan took place on Day 1 before the remote learning session, meaning that the change score from pre- to post-learning was computed across 24 hr for the recent condition but within the same scan session for the remote condition. We were concerned that this difference may insert noise into the change score for the recent condition. We reasoned that if this were the case, changes in coupling from pre-learning to post-recent learning rest may be more variable than changes in coupling from pre-learning to

post-remote learning rest. We conducted *F*-tests to compare the variance of the change in these two hippocampal–LOC correlations and found no reliable difference (ratio of difference: $F_{(22, 22)} = 0.811$, p = 0.63), suggesting that the change score for the recent condition was not substantially noisier than that of the remote condition.

As an additional precaution, we side-stepped this potential confound of a difference in the change scores by computing the correlation between hippocampal–LOC coupling and priming separately for the pre- and post-learning scans (*Figure 4—figure supplement 1*). We found that neither pre- nor post-learning coupling related to priming of recently learned sequences (pre-learning: $r_{(21)} = -0.19$, p = 0.38, post-learning: $r_{(21)} = -0.07$, p = 0.75). Notably, the finding that the relationship between change in connectivity and priming of remotely learned sequences was driven by trends for a negative relationship pre-learning ($r_{(21)} = -0.40$, p = 0.06) and a positive relationship post-learning ($r_{(21)} = 0.39$, p = 0.07).

These findings suggest two things. First, the lack of a relationship between connectivity and priming of recently learned sequences is not due to a noisier change score that was computed across sessions, as there was no relationship with priming when isolating the post-learning rest scan. In contrast, the relationship between coupling and priming of remotely learned sequences was observed in the post-learning scan alone, albeit as a statistical trend. Second, these findings highlight the importance of employing a change score, as we observed a trend for a negative relationship between coupling and priming before learning occurred. This may be spurious, or it could reflect individual differences in intrinsic brain function (*Fox et al., 2007*), but regardless needs to be accounted for to identify signals that are selective to learning and consolidation processes.

## Experiment 2: relationship between neural integration and rest connectivity

So far, we have reported that priming of overlapping sequences is related both to changes in their neural similarity in mPFC and increases in rest connectivity between posterior hippocampus and LOC. In the next section, we aimed to examine the relationship between these two neural measures and test whether they contributed unique variance in the extent of recognition priming observed across participants.

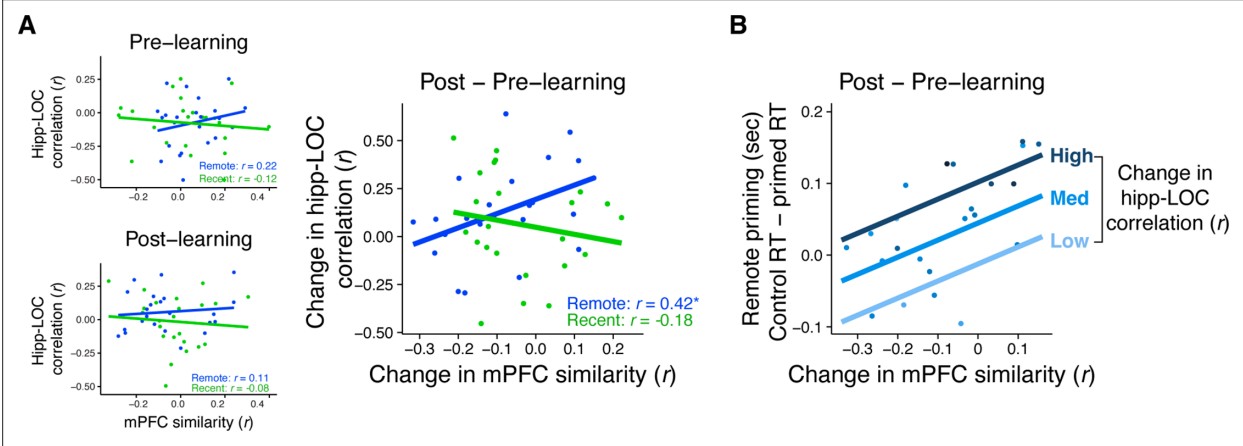

**Figure 5.** Relationship between mPFC similarity, rest connectivity, and priming. (**A**) Correlation between medial prefrontal cortex (mPFC) similarity and rest connectivity between posterior hippocampus and LOC across participants, separately for pre- and post-learning connectivity and pre- and post-learning similarity in mPFC (left) and for the change in connectivity and similarity (post- minus pre-learning; right). Dots represent participants and lines represent model fit. * indicates p < 0.05. (**B**) Visualization of multiple linear regression predicting recognition priming for the remotely learned sequences. Each dot is a participant. Color indicates the magnitude of the change in rest connectivity between posterior hippocampus and LOC. Lines represent fits of the model at different magnitudes of rest connectivity. Plots can be reproduced with *Figure 5—source data 1–3*.

The online version of this article includes the following source data for figure 5:

**Source data 1.** Relationship between mPFC similarity and rest connectivity, separately for pre- and post-learning rest scans.

**Source data 2.** Relationship between change in mPFC similarity and change in rest connectivity.

**Source data 3.** Relationship between mPFC similarity, rest similarity, and recognition priming.

## Relationship between similarity and rest

The relationship between mPFC similarity and hippocampal–LOC rest connectivity was computed by correlating the two measures across participants separately for the recent and remote learning sessions (*Figure 5A*, right). We found that participants with a greater increase in mPFC similarity of overlapping sequences also exhibited a greater increase in rest connectivity immediately post-learning, but only for objects from the remote session ($r_{(21)} = 0.42$, p = 0.048) and not the recent session ($r_{(21)} = -0.18$, p = 0.40). Furthermore, this relationship was reliably stronger for the remote session over the recent session ($z = 2.04$, p = 0.04). Critically, when examining the relationship between rest connectivity with mPFC similarity separately using pre- and post-learning measures of each variable, no correlations reach statistical significance (all *r*'s < 0.22, all p's > 0.31; *Figure 5A*, left), which further underscores the importance of focusing on learning-related changes in these measures.

## Neural measures related to priming

Since the neural measures of mPFC similarity and LOC–hippocampal rest connectivity were correlated for the remotely learned sequences, we next assessed whether the two measures explained unique or shared variance in their relationship with recognition priming. Focusing on data from the remote learning session, we computed a multiple regression with recognition priming (i.e. the median differences in primed versus unprimed RTs) as the dependent variable, and two predictors: the change in rest connectivity and the average change in mPFC similarity. We then computed partial $R^2$ values for each predictor. This revealed unique contributions of both the change in mPFC similarity ($t_{(1, 20)} = 2.37$, p = 0.03, semi-partial $R^2 = 0.39$) and the change in posterior hippocampal–LOC rest connectivity ($t_{(1, 20)} = 2.13$, p = 0.045, semi-partial $R^2 = 0.43$) in explaining recognition priming across participants. *Figure 5B* visualizes the results of this model, showing participants' relationship between their recognition priming of the remotely learned objects and their change in mPFC similarity. This relationship is overlaid with lines of best fit that represent the relationship between mPFC similarity and priming given three magnitudes of hippocampal–LOC connectivity. This suggests that both measures are *positively* and *uniquely* associated with participants' time-dependent integration across overlapping sequences.

# Discussion

Across two experiments, we investigated how cortical representations and post-learning coupling influenced the behavioral integration of memories with overlapping predictive structure. We manipulated overlap in sequences of triplets such that objects shared antecedents (i.e. both were predicted by the appearance of the same pair of objects) but never occurred together in time. First, we found that participants' RTs reflected increased association of objects with shared temporal structure, but only 24 hr after learning the sequences. This delay-dependent behavioral measure of integration was replicated in a cohort that underwent fMRI. In this cohort, we found that cortical neural similarity was shaped by sequence learning in a delay-dependent manner: patterns of activity in LOC reflected the immediate association of objects from overlapping sequences, while mPFC differentiated objects from overlapping sequences learned 24 hr prior. At the same time, learning of the sequences increased post-learning connectivity between the posterior hippocampus and LOC. Critically, after a 24-hr delay, changes in mPFC similarity and changes in hippocampal–LOC rest connectivity were correlated across participants, and both measures explained unique variance in the extent of recognition priming across participants. We interpret these findings as evidence that both coordinated hippocampal–cortical coupling and cortical learning are markers of consolidation that contribute to the behavioral integration of overlapping memories over time. Below, we discuss each of these findings through the lens of systems consolidation theories and in the context of prior empirical work.

## Integration of overlapping regularities

As a reminder, we observed implicit integration across overlapping sequences in a recognition priming protocol, only after a 24-hr period, suggesting that systems-level consolidation processes can enhance behavioral integration across memories with overlapping content. This finding adds to the literature showing consolidation-dependent memory integration in different tests of integration: in transitive inference (*Ellenbogen et al., 2007*; *Lau et al., 2010*; *Werchan and Gómez, 2013*), the extraction of

statistical regularities (*Wagner et al., 2004*; *Durrant et al., 2011*; *Durrant et al., 2011*; *Sweegers et al., 2014*; *Batterink and Paller, 2017*) and category learning (*Djonlagic et al., 2009*; *Graveline and Wamsley, 2017*). The observed integration seems to emerge over time due to a slowing of the control items, rather than a facilitation of the primed items. It is important to interpret this in light of the recognition decisions that participants made as a cover task. Slowing of recognition responses over delay periods is a classic time-dependent effect (*Reber et al., 1997*), and the fact that the RTs in the primed condition do not show this slowing suggests that implicit integration served as a protection against the slowing that would otherwise occur. However, as we did not find statistically reliable interaction between priming condition and day in either experiment, the notion that such integration develops solely after a 24-hr delay should be evaluated with caution.

While we only observed behavioral priming 24 hr after learning, we observed weak but reliable explicit integration for the same items immediately after learning, which remained unchanged over the same 24-hr period, consistent with many observations showing immediate explicit integration across overlapping memories (*Acuna et al., 2002a*; *Acuna et al., 2002b*; *Ellenbogen et al., 2007*; *Greene et al., 2006*; *Heckers et al., 2004*; *Lau et al., 2010*; *Preston et al., 2004*; *Werchan and Gómez, 2013*; *Zeithamova et al., 2012*). Even though both results are consistent with prior work, what could explain the discrepancy in the timing of successful implicit and explicit integration—both of which replicated in a second cohort of participants? A handful of behavioral studies find that participants perform better at explicit integration tests after sleep, but these studies differ in many factors that could explain the discrepancy with our findings: (1) instructions to explicitly encode premise pairs (*Ellenbogen et al., 2007*), (2) foreknowledge of the underlying structure of the stimulus associations prior to learning (*Sweegers and Talamini, 2014*), and (3) the use of non-temporal versus temporal structure of the regularities (*Lerner and Gluck, 2019*).

The paradigms that specifically investigate the integration of sequences merit special attention. There is some evidence of transitive inference across overlapping sequences (e.g. inferring AC from viewing AB and BC in a continuous stream of colored dots) (*Luo and Zhao, 2018*). Explicit recognition of these associations was on par with performance in our explicit integration task. However, there was no delayed test or implicit integration measure, precluding any interpretations about the role of consolidation in this form of integration. The sequences in our protocol also resemble paradigmatic relations (*McNeill, 1963*), which are second-order associations that do not co-occur, but instead are substitutable with each other because of their shared context (e.g. wearing 'flip-flops' and wearing 'boots'). Interestingly, *Yim et al., 2019* observe implicit learning of paradigmatic relations only when participants are instructed to attend to the stimuli (i.e. actively categorizing images rather than listening to an auditory stream while coloring) and have robustly learned the relevant premise pairs. This fits with our findings in several ways: first, our encoding task required constant comparisons between the present and previous objects, which may have heightened awareness of the temporal regularities, like in their active encoding protocol. Second, Yim et al. find strong evidence of explicit memory for first-order associations across all experiments, which mirrors our observations of strong explicit knowledge of AB pairs. However, one notable difference is that Yim et al. did not administer an implicit test for paradigmatic associations after a delay, which is the only condition where we observe integration. Why integration emerges immediately after learning in their experiment, but only after a period of consolidation in our experiment, remains an open question.

Finally, the discrepancy between our implicit and explicit tasks could be explained more broadly by differences in memory strength for the within- and across-event associations. Results from the explicit memory test indicate much better integration of objects in the same sequence (A and B) relative to integration across sequences ($C_1$ and $C_2$). Importantly, A and B were viewed twice as many times as each C, as they preceded each appearance of $C_1$ and $C_2$. It may be that these overlearned associations are more accessible and explicit integration tests are sensitive to this information. At the same time, consolidation may promote representational changes that (1) benefit associations that are not already strengthened through sufficient learning (*Schapiro et al., 2018*), or (2) benefit implicit memories, which are more easily captured with measures like priming, more than explicit, declarative memories (*Henke, 2010*). These possibilities also fit well with TTT, which posits that differences in how a memory is expressed may be explained both by the relative strength of different neural traces at different time points, and by which of multiple neural traces is most suitable for the demands of the current task. It may be that neural representations supporting implicit integration may require time and consolidation

to develop, while neural representations that can leverage explicit retrieval strategies may be available immediately after learning. The use of multiple behavioral tasks that capture both explicit and implicit expressions of memories is a fruitful approach that could be used to better characterize how different memory traces may be differentially expressed over time.

While we consider our priming task to be an implicit test of integration across sequences, the task does rely on explicit memory demands, as responses in the priming task reflect explicit recognition of each object in isolation. The use of a recognition task raises the question of whether our results would generalize to a task that does not rely on memory retrieval. Recently, there have been observations of implicit integration using a similar priming task in which participants made preference ratings for the learned stimuli (*Abolghasem et al., 2023*). Given this, we speculate that response priming can reveal newly learned associations between stimuli regardless of the cover task. On the other hand, responses in the learning task required an explicit comparison of two objects in sequence, as participants decided if the object on screen was bigger or smaller than the preceding one. This emphasis on the temporal relationships between objects may have quickened participants' ability to learn the sequences and in turn facilitated integration across overlapping sequences. Without this attention to temporal relationships, we speculate that learning of the sequences still may have occurred, but more exposure to the sequences would be required to reach the same magnitude of learning.

What learning mechanism could enable the integration of overlapping temporal regularities? One candidate is forward prediction: when encountering repeated sequences of events, neural representations of the predicted items are formed and strengthened over subsequent repetitions of the sequence (*Kok et al., 2012*; *Schapiro et al., 2012*; *Turk-Browne et al., 2012*; *Schapiro et al., 2013*; *Hindy et al., 2016*; *Schapiro et al., 2016*; *Kok and Turk-Browne, 2018*; for a review of hippocampal involvement, see *Davachi and DuBrow, 2015*). In our case, due to the intermixed exposure to the same AB and two different C objects, forward prediction of both C objects while viewing B may have helped to cement their association. More specifically, the early comparisons between the two outcomes of an overlapping sequence may have produced a prediction error that directed attention to the differences across the two sequences, which would give rise to their successful retention and integration (*Wahlheim and Zacks, 2019*). Notably, the frequent interleaved repetition of the two overlapping sequences may have additionally strengthened their association; without this interleaved training, overlapping sequences may have been behaviorally differentiated in order to reduce interference across them (*Chanales et al., 2020*; *Drascher and Kuhl, 2022*). Forward prediction may not be the only mechanism that can give rise to the integration, as there is neural evidence that second-order associations built with predictable first-order associations are reflected in patterns of brain activity, suggesting that they can be learned in the absence of predictive mechanisms (*Schapiro et al., 2013*; *Schapiro et al., 2016*). Furthermore, the fact that we only observed behavioral integration after a delay suggests that forward prediction may not be sufficient to give rise to behavioral integration across events.

## Cortical and hippocampal similarity

We found that after 24 hr, patterns of activation in mPFC reflected differentiation of objects from overlapping sequences. This was counter to our prediction that with consolidation, neural patterns in mPFC would reflect similarities across sequences with shared temporal regularities, as most evidence of neural differentiation of highly similar or overlapping stimuli is observed in the hippocampus rather than in cortical regions (*Hulbert and Norman, 2015*; *Favila et al., 2016*; *Chanales et al., 2017*; *Dimsdale-Zucker et al., 2018*). There are a few noteworthy findings of mPFC differentiation that may help to make sense of this finding. First, *Ezzyat et al., 2018* observed that neural patterns evoked by associative memories in mPFC were more distinct than those evoked by item memories, but this difference only held for events that were re-studied after a night of sleep and not for events re-studied in the same experimental session. Second, individual autobiographical memories can be successfully decoded in mPFC, and classification accuracy for individual memories is greater for more remote memories relative to recent ones (*Bonnici et al., 2012*). Third, differentiation in mPFC has been shown to emerge over repeated testing, in addition to over a delay period, and also relates to long-term memory (*Karlsson Wirebring et al., 2015*), in line with the idea that repeated testing can accelerate the stabilization of memories by minimizing competition with related memories (*Antony et al., 2017*; *Hulbert and Norman, 2015*). Fourth, different regions of mPFC exhibit integration and

differentiation signatures within the same experimental setting (*Schlichting et al., 2015*), suggesting that there may be anatomical distinctions across this region that were not considered in the present experiment. Nevertheless, we suggest that mPFC plays a role in the differentiation of discrete memories to support their long-term storage.

Perhaps the most surprising results we found were that despite this *differentiation* in mPFC when considering group-level changes, mPFC neural patterns reflecting *integration* positively scaled with priming across participants. This suggests that the level of neural representational overlap in the same subregions of mPFC may support both differentiation and integration of overlapping memories— extending beyond findings that different subregions within mPFC separately support differentiation and integration (*Schlichting et al., 2015*). Reports of correlations between neural and behavioral integration despite neural differentiation at the group level have been reported before (*Molitor et al., 2021*). This suggests that the restructuring of memories in mPFC conforms to ongoing goal states in addition to the properties of the experience itself (e.g. whether there are similarities or differences across events). Consistent with this, in past work, we found that neural pattern overlap increased in mPFC for objects paired with the same scene after a week, but these effects were observed in a source memory test focusing on the scenes, which either overlapped or were distinct from other memories, and less so in a standard recognition test of the experiment-unique objects (*Tompary and Davachi, 2017*). Furthermore, in a similar experiment where participants were required to make more fine-grained source memory judgements, mPFC patterns only reflected increasing neural integration over time when the encoded object–scene pairs were congruent with prior knowledge (*Audrain and McAndrews, 2022*). These mixed results suggest that task demands matter: they likely prioritize processes that are most needed for successful performance (*Brunec et al., 2020*), which could give rise to differentiated or integrated neural patterns evoked by the same stimuli depending on the task. Taken together, however, the extant data combined with past results supports the conclusion that dynamic neural representational change occurs in the mPFC that structures our experiences, but that it may be best to think of these representations as lying on a continuum from separated to integrated, and thus able to support behavior across a variety of tasks.

In this study, the pattern of activity in the anterior hippocampus reflected learned associations between objects that always appeared back-to-back (A and B). This is a direct replication of prior work (*Schapiro et al., 2012*) and also highlights the importance of hippocampal representations in memory for temporal order (*DuBrow and Davachi, 2013*; *Hsieh et al., 2014*; *Kalm et al., 2013*; *Paz et al., 2010*). At the same time, there are several reports of hippocampal integration across events that are separated in time, both immediately after learning (*Ritchey et al., 2015*; *Schapiro et al., 2016*, p. 201; *Schlichting et al., 2015*) and after a delay (*Dandolo and Schwabe, 2018*; *Ritchey et al., 2015*; *Tompary and Davachi, 2017*), and even lesion work demonstrating the necessity of the hippocampus for such integration behaviors (*Pajkert et al., 2017*; *Schapiro et al., 2014*). Why then did we not observe changes in neural similarity in the hippocampus that reflected overlap across sequences separated in time? One possibility is that any signal reflecting overlap in the hippocampus may have been too subtle to identify in the exposure phase, as participants were not engaging with the objects in a manner that would promote integration processes. In contrast, signals reflecting AB integration may have been strong enough to appear in the absence of an active integration task because AB sequences were viewed twice as many times and were strongly explicitly remembered. Furthermore, as with any null effect, it could be due to low statistical power or low signal-to-noise ratio, a common issue for MRI investigations of medial temporal lobe structures.

## Post-encoding rest connectivity

A critical component of systems-level consolidation theories is that long-term memories become stabilized through communication between the hippocampus and cortex, resulting in cortical memory traces. Here, we found that learning object sequences resulted in increased rest connectivity between the posterior hippocampus and LOC, a region that codes for object information. This increase in connectivity was related to the extent of recognition priming across participants, suggesting that coordinated post-learning processing in these regions can facilitate the behavioral integration of overlapping memories. This finding extends a growing body of work demonstrating that hippocampal connectivity with stimulus-selective cortex is related to long-term memory retention (*Collins and Dickerson, 2019*; *de Voogd et al., 2016*; *Keller and Just, 2016*; *Murty et al., 2017*; *Murty et al., 2019*).

However, our findings differ from these observations in one critical way: we found that post-learning rest connectivity related to behavioral integration across overlapping memories, rather than retention of discrete memories. To our knowledge, the only one analogous finding has been observed, in a paradigm where post-learning rest connectivity between anterior hippocampus and the fusiform face area was measured after learning premise pairs in an associative inference paradigm (*Schlichting and Preston, 2016*). Together this suggests that post-learning rest connectivity may not only support the retention of specific episodic memories, but also underpin integration across them. When might consolidation processes support one over the other? One possibility could be that salient features surrounding learning shape consolidation to prioritize goal-relevant information (i.e. *Cowan et al., 2021*). However, in our study and the one conducted by Schlichting and Preston, the integration tests were a surprise. Without cues about the test, it remains unclear which feature of memory may be prioritized by consolidation mechanisms, or if both are. Future studies should relate learning-related rest connectivity with distinct behavioral measures that capture both retention of individual events and across-event integration.

We also identified a delay-dependent relationship between changes in rest connectivity and cortical integration of objects from overlapping sequences. Specifically, participants with elevated posterior hippocampal–LOC rest connectivity after the remote learning session also exhibited greater neural integration in mPFC for the remotely studied objects. Both measures explained variance in recognition priming across participants, suggesting that they uniquely support delay-dependent behavioral integration. The timing of these two measures may hint at the directionality of their association: the rest scan occurred immediately after learning on Day 1, and the post-learning similarity values were measured on Day 2. This order parallels predictions from systems-level consolidation theories that cortical neural traces are trained by the hippocampus via coordinated processing (*McClelland et al., 1995*). An intuitive prediction is that coordination between the hippocampus and a particular cortical region would shape similarity in that same region. Indeed, in one study, pattern similarity in mPFC for objects studied in the same temporal context was related to connectivity between the anterior hippocampus and mPFC (*Cowan et al., 2020*). Relatedly, past work has found that connectivity between anterior hippocampus and mPFC was related to the similarity of overlapping memories in mPC (*Cowan et al., 2020*) and anterior hippocampus (*Tompary and Davachi, 2017*). In other words, both studies identified the same circuit despite differences in the site of representational change and in their experimental designs. However, the current findings suggest that the connection between post-learning connectivity and similarity is more complex, since we found that changes in similarity in mPFC related to changes in posterior hippocampal connectivity with LOC and not anterior hippocampal connectivity with mPFC. This is especially surprising when considering past work that more often implicates anterior hippocampus in memory integration (*Schapiro et al., 2012*; *Schlichting et al., 2015*; *Ritchey et al., 2015*). One possibility for this discrepancy is that the anterior hippocampus may support online and/or explicit integration, while the role of posterior hippocampus is to strengthen memory for discrete items in offline periods that can later be integrated via the anterior hippocampus. However, this is difficult to reconcile with the above-mentioned demonstration that offline connectivity between anterior hippocampus and mPFC relates to neural integration (*Tompary and Davachi, 2017*; *Cowan et al., 2020*). Further investigation of relationships between cortical similarity and post-learning processing is needed to constrain theories of systems-level consolidation and understand the precise circuit interactions unfolding over time that give rise to consolidated memories.

## Future directions, caveats, and conclusion

There are two major topics that we have so far neglected to mention but are integral to the study of the consolidation and integration of related events. First, new memories are not learned and consolidated in a vacuum. Prior knowledge has long been known to influence the formation of new event memories (*Alba and Hasher, 1983*; *Anderson, 1984*; *Bartlett, 1932*; *Bransford and Johnson, 1972*). More recently, much work has been devoted to understanding how prior knowledge influences neural processes that support their encoding, long-term storage, and transformation over time (*Liu et al., 2017*; *Liu et al., 2018*; *Bonasia et al., 2018*; *Bellana et al., 2021*; *Audrain and McAndrews, 2022*). In our experiment, all object stimuli were real-world objects, thus the random assignments of object-triplet across participants may have incidentally introduce uneven pre-experimental associations among the sequences (e.g. belonging to the same category, or having similar functions, or sharing

salient features). This raises interesting questions about how prior knowledge may influence the integration of new related memories in addition to their long-term stabilization, a promising avenue of future work.

Second, TTT is agnostic to whether consolidation mechanisms require the passage of time, or if there are circumstances in which integrated, cortically based memories can form quickly. There are now many demonstrations that memories can be rapidly consolidated, as characterized by longer retention durations and increased involvement of mPFC: through repeated retrieval (*Antony et al., 2017*; *Ferreira et al., 2019*; *Ye et al., 2020*), through 'fast-mapping' of novel words onto new concepts (*Coutanche and Thompson-Schill, 2015*; *Sharon et al., 2011*); and for events that are consistent with prior knowledge, as discussed above. Furthermore, discrete events can be integrated into more general knowledge, like schemas, immediately after learning (*Brown and Evans, 1969*; *Kumaran et al., 2009*; *Posner and Keele, 1968*; *Richter et al., 2019*; *Tompary et al., 2020*; *Tompary and Thompson-Schill, 2021*; *Zeng et al., 2021*). However, this does not directly conflict with TTT, as new schemas are increasingly used after a delay even though they are immediately available (*Tompary et al., 2020*; *Zeng et al., 2021*), consistent with TTT's prediction that the form of memory that is expressed (detailed or schematic) is governed by the relative strength of the neural trace of that memory in the hippocampus versus cortex. Future work should test whether the consolidation-related neural measures presented in this study may similarly underpin the rapid integration of new information.

Finally, we note several important caveats. First, to manipulate the passage of time/course of consolidation within participants, we split encoding into two sessions and intermixed encoded memoranda into one test session. There are several advantages and disadvantages to this design over a design in which with one encoding session and retrieval split into 'immediate' and 'delayed' sessions. In our design, participants are only tested once, preventing rehearsal of stimuli to be tested after a delay after exposure to the test format in the 'immediate' session. An intermixed test also minimizes the chances that participants are switching retrieval strategies or response criteria for recent and remotely learned memoranda. However, encoding in the second 'recent' session may be influenced by the first 'remote' session. Indeed, knowledge of the task is reflected in the overall slower RTs but a larger facilitation in RTs across learning in the remote session relative to the recent one (*Figure 1—figure supplement 1*). As we discussed, it is well known that prior knowledge can shape the encoding of new, similar events, often with a shift to circuits involving mPFC when encoding of events that are consistent with prior knowledge (e.g. *van Kesteren et al., 2010*). This may explain why we find increased rest connectivity of LOC and posterior hippocampus after the first, 'remote' session but increased rest connectivity of LOC and mPFC after the second, 'recent' session; both observations may be driven by processing of the remote memories, reflecting a shift from hippocampal to cortical processing across the two sessions. Indeed, the post-learning rest scan from the 'recent' scan may reflect processing of both recently and remotely learned stimuli, and this mixture of signals may have prevented us from identifying a clear link between rest connectivity and memory. In summary, neither a design with split encoding nor a design with split retrieval can perfectly isolate influences of consolidation. Ideally, in the future, consolidation effects will be observed in converging findings from both types of designs.

A second caveat is that we did not include a pre-learning rest scan during the 'recent' session. We chose not to include such a scan for several reasons. First, the Day 2 session was longer than that of Day 1 because it included the recognition priming and explicit memory tasks, and the addition of a pre-learning scan would have made the length of the session longer and more tiring for participants. Second, because we split learning into two sessions as discussed above, we anticipated that the pre-learning scan would not have been a 'clean' measure of baseline processing, but rather would include signal related to continued processing of the Day 1 'remote' sequences, as multivariate reactivation of learned stimuli has been observed in rest scans collected 24 hr after learning (*Schlichting and Preston, 2014*). For these reasons, we decided to use the pre-learning rest scan from the Day 1 'remote' session as a baseline for both learning sessions. However, there is a concern that using a pre-learning scan from a different day might generated a noisier estimate of a change score between that scan and the post-recent learning scan, as patterns of resting-state connectivity are more correlated across two scans within the same experimental session relative to across two sessions (*Shehzad et al., 2009*). This added noise may have prevented us from observing a change in

hippocampal–LOC coupling after recent learning, which would be equivalent to what we observed for remotely learned sequences and would reflect immediate offline processing of the recently learned sequences. However, as stated before, we instead observed a reliable increase in coupling between LOC and mPFC, which suggests that a change score computed across scan sessions is sensitive enough to identify changes in coupling at the group level. Furthermore, the fact that there was no relationship between a change in coupling and recognition priming of recent objects may be driven by the facts that there was no reliable evidence of priming of recently learned sequences in the recognition task.

A third caveat is that the post-learning exposure phase always followed recognition priming, which was the only task in which participants viewed $C_1$ and $C_2$ objects back-to-back. This new sequential order may have impacted their corresponding neural representations. When designing the study, we reasoned that it was more important for the behavioral priming task to come before the exposure scans, as all objects were shown only once in that task, whereas they were shown four to five times in a random order in the post-learning exposure phase. Because of this difference in presentation times, and because behavioral priming findings tend to be very sensitive, we reasoned that it was more important to protect the priming task from the exposure scan instead of the reverse. Furthermore, we expected that the single additional presentation of the C objects in the recognition priming task would not substantially override their sequence learning, as C objects were each presented 16 times in their sequence (ABC$_1$ and ABC$_2$ 16 times each). Critically, the order of C objects during recognition was the same for recent and remote conditions; since we observed a selective change in neural representation for the remote condition and no corresponding change for the recent condition, this suggests that recognition priming order alone could not substantially impact the representations.

A fourth caveat concerns the control condition in the priming task. The control condition comprised responses to C objects that followed baseline items rather than C objects from different sequences. We decided to use baseline items as we wished to maximize the number of comparisons in the primed condition for power purposes while only presenting each object once—as mentioned above, repetition of objects in this task may have further facilitated RTs and potentially masked any priming effects. However, this design choice leaves opens the possibility that *all* C objects became behaviorally integrated regardless of their sequence overlap, due to some other dimension of the learning task. For instance, all C objects are positioned 'third' in each sequence, and they have identical transition probabilities. We find encouraging evidence that such integration by position is unlikely. First, the observed priming effect for the remotely learned sequences is selectively correlated with post-learning neural similarity between C objects from overlapping sequences, and not related to similarity between C objects from different sequences (*Figure 3—figure supplement 1*). Furthermore, the explicit integration tests in both experiments used C objects from other sequences as foils, either appearing in re-arranged pairs in Experiment 1 or as the foil option in Experiment 2. Above-chance performance in both experiments suggests that participants successfully learned some sequence-specific associations, rather than merely learning positional information about the different objects. However, future work with a sequence-specific control condition would strengthen our claim that priming between the C objects specifically reflects the integration of objects from overlapping events.

Taken together, the present results demonstrate that over time, episodic memories that share temporal regularities become behaviorally integrated. Furthermore, these findings provide evidence for the notion that psychological transformations in memories are accompanied by shifts in cortical similarity—both integration and differentiation—and driven by post-learning hippocampal–cortical coupling. These observations reveal new insights into the development of conceptual or semantic memory over time, suggesting that the consolidation of memories with overlapping temporal structures may constitute a key driver of memory integration.

## Materials and methods
### Experiment 1
#### Subjects
Ninety right-handed, native English speakers participated in this experiment. Demographic information was lost for 25 participants; of the 65 with intact records, 42 were female, and the mean age was 23 (range: 18–33). Participants were recruited from New York University and the broader community. The University Committee on Activities Involving Human Subjects approved all recruitment and

consent protocols. Twelve participants were excluded: six did not return for the second experimental session; one fell asleep; five ended early due to experimental error. A further 5 participants were excluded due to poor performance on the recognition test ($A < 3$ SD relative to group average for either recent or remote memory), leaving 73 participants included in all following analyses.

## Stimuli
The stimuli used in this experiment consisted of 100 color images of objects, taken from online databases and used in previous studies (*DuBrow and Davachi, 2013*; *DuBrow and Davachi, 2014*).

## Experiment procedure
The experiment consisted of two sessions, separated by approximately 24 hr. In the first session, participants incidentally encoded sequences of the objects while performing a cover task (remote learning). In the second session, participants began by incidentally learning a new set of object sequences while performing the same cover task as in the first session (recent learning). Then, participants completed two tests consisting of both sets of stimuli from the two learning sessions: a recognition priming task and an explicit integration task. Because memory for all object sequences was tested at the end of the second scan session, memory for objects learned on the first day were considered remote, and memory for objects learned on the second day were considered recent. Thus we refer to the first session as the *remote* session and the second session as the *recent* session.

## Sequence learning task
In both sessions, participants incidentally encoded a set of sequences. As a cover task, participants viewed each object and decided if the object present on screen was bigger or smaller than the prior object. All objects were presented as the same size, but participants were instructed to use estimates of their real-life sizes.

The order of the objects was arranged such that the first two objects in a sequence (A and B) were presented together 100% of the time. B was directly followed by one of two other objects ($C_1$ and $C_2$), each following B 50% of the time. Thus, participants were exposed to $ABC_1$ half of the time, and $ABC_2$ the other half of the time. Six unique sequences of objects were presented in each of the two learning sessions. Six baseline objects were randomly inserted between sequences, for use as control comparisons for C objects (see Methods for priming task below). This resulted in a total of 30 objects in each learning session: 6 A objects, 6 B objects, 6 $C_1$ objects, 6 $C_2$ objects, and 6 baseline objects. See *Table 1A* for transition probabilities between all objects.

Each sequence of $ABC_1$ and $ABC_2$ was repeated 16 times, meaning that A and B were exposed to participants 32 times and $C_1$ and $C_2$ were exposed to participants 16 times over the course of learning. Baseline objects were presented 16 times to equate exposure frequency with C objects. In other words, $ABC_1$ and $ABC_2$ sequences would be intermixed either with baseline objects or other sequences. The presentation of all sequences and baseline objects were divided into four 11-min learning blocks per session, and participants were given the option to take a 1-min break between blocks. The order of the stimuli was pseudo-randomized such that all sequences ($ABC_1$ and $ABC_2$) appeared four times in each block, all baseline objects appeared twice in each block, and no sequence with the same A and B objects was presented back-to-back.

The presentation timing was designed in anticipation of the planned fMRI experiment (see Experiment 2). Each object was presented for 2 s, and participants were instructed to respond as quickly as possible before the object was removed from the screen, without sacrificing accuracy. Between each trial, there was a variable inter-trial interval (ITI) ranging from 0 to 4 s. The ITI was constructed to optimize item-level pattern similarity analyses between $C_1$ and $C_2$ within a sequence (not reported in this manuscript), while also allowing for sufficient jittering of trials to perform condition-level univariate analyses. The ITIs between sequences and before and after baseline objects were randomly jittered.

Finally, the sizes of objects in each sequence were pre-determined such that the motor response to C objects were matched within each sequence. To do this, we divided the object into 'big' and 'small' bins based on estimations of their size relative to a shoe box. We then arranged the objects into sequences such that both $C_1$ and $C_2$ were either bigger or smaller than B. Furthermore, in half of the sequences in each learning session, $C_1$ and $C_2$ were bigger than B, and in the other half, $C_1$ and $C_2$ were smaller than B. However, because the sequences and baseline object were randomly intermixed,

**Table 1.** Mean (SD) transition probabilities for all objects during.
(A) Learning in both experiments, (B) recognition priming in both experiments, (C) the first exposure scan, and (D) the second exposure scan. As a reminder, the pre- and post-learning exposure phases were divided into two scans, one that comprised A, $C_1$, and baseline objects, and one that only comprised B, $C_2$, and baseline objects (see Methods). Italicized values indicate transition probabilities that additionally are specific to objects from the same sequence. For example, during learning, not only does a B object follow every A object, but the same B object follows the same A object 100% of the time. In contrast, baseline objects are always followed by A objects, but the specific objects vary.

| A | | Follows | | | | | |
|---|---|---|---|---|---|---|---|
| | | A | B | $C_1$ | $C_2$ | Base | |
| | A | 0 (0) | *1 (0)* | 0 (0) | 0 (0) | 0 (0) | |
| | B | 0 (0) | 0 (0) | *0.5 (0)* | *0.5 (0)* | 0 (0) | |
| Precedes | $C_1$ | 0.5 (0.04) | 0 (0) | 0 (0) | 0 (0) | 0.5 (0.04) | |
| | $C_2$ | 0.5 (0.04) | 0 (0) | 0 (0) | 0 (0) | 0.5 (0.04) | |
| | Base | 1 (0) | 0 (0) | 0 (0) | 0 (0) | 0 (0) | |
| **B** | | Follows | | | | | |
| | | A | B | $C_1$ | $C_2$ | Base | Foil |
| | A | 0 (0) | *1 (0)* | 0 (0) | 0 (0) | 0 (0) | 0 (0) |
| | B | 0.11 (0.08) | 0 (0) | 0 (0) | 0 (0) | 0.18 (0.10) | 0.71 (0.11) |
| | $C_1$ | 0 (0) | 0 (0) | 0 (0) | *1 (0)* | 0 (0) | 0 (0) |
| Precedes | $C_2$ | 0.15 (0.10) | 0 (0) | 0 (0) | 0 (0) | 0 (0) | 0.84 (10) |
| | Base | 0 (0) | 0 (0) | 1 (0) | 0 (0) | 0 (0) | 0 (0) |
| | Foil | 0.22 (0.03) | 0 (0) | 0 (0) | 0 (0) | 0.25 (0.03) | 0.52 (0.03) |
| **C** | | Follows | | | | | |
| | | A | $C_1$ | Base | | | |
| | A | 0.39 (0.05) | 0.40 (0.06) | 0.21 (0.04) | | | |
| Precedes | $C_1$ | 0.39 (0.06) | 0.42 (0.05) | 0.19 (0.05) | | | |
| | Base | 0.43 (0.08) | 0.37 (0.07) | 0.20 (0.06) | | | |
| **D** | | Follows | | | | | |
| | | B | $C_2$ | Base | | | |
| | B | 0.39 (0.05) | 0.39 (0.05) | 0.21 (0.04) | | | |
| Precedes | $C_2$ | 0.41 (0.04) | 0.39 (0.05) | 0.21 (0.04) | | | |
| | Base | 0.40 (0.08) | 0.44 (0.08) | 0.16 (0.05) | | | |

motor responses to A, B, and baseline object were not matched in this same manner (e.g. a 'big' baseline object could precede a 'big' A object). Because of this, accuracy of size judgments could not be derived for these comparisons because the objects' relative sizes might be judged differently according to participants' preferences and past experiences. Thus, accuracy was reported only for the images with a clear answer (e.g. a 'big' object following a 'small' object or vice versa), and RTs were reported for all images regardless of accuracy for the objects with a clear answer.

## Recognition priming task

In this task, all 60 objects from both learning sessions were presented intermixed with 40 novel foils. As a cover task, participants were instructed to endorse each object as 'old' or 'new'. They were asked

to respond as quickly as possible without sacrificing accuracy, and they were not given an opt-out or 'don't know' option. The ITI between each trial was fixed at 1 s.

Unbeknownst to the participants, the order of the objects was manipulated to use response priming as an index of participant's implicit knowledge of the association between $C_1$ and $C_2$. A priming manipulation was used to test memory for 'across-episode' associations: namely, the association between C objects that followed the same sequence of A and B. Critically, $ABC_1$ and $ABC_2$ were never experienced together, but through the overlap in A and B objects, we expect the two episodes to become associated. Note that to participants, $C_1$ and $C_2$ objects hold an identical mnemonic status, as they both follow B an equal number of times and are presented within their sequence in randomized order over learning. We have assigned them with separate labels for the sole purpose of clarifying the conditions of the priming manipulation. If participants successfully associated $C_1$ and $C_2$ through their shared prior sequential information, we expected that the presentation of $C_1$ before $C_2$ would facilitate processing of $C_2$, resulting in a faster RT. Each $C_1$ object directly preceded the $C_2$ object that shared an overlapping sequence of A and B during learning. As a control comparison, each $C_1$ object followed a baseline object learned in the same session. Since C and baseline objects share no sequential information, we predicted that the baseline object would not facilitate of processing of C. This order (baseline, $C_1$, and $C_2$) also controlled for motor response history, as we only included objects that (1) were correctly endorsed as 'old' and (2) whose prior objects were also correctly endorsed as 'old'. See *Table 1B* for transition probabilities between all objects during the recognition task.

Notably, the sequences that comprise both primed ($C_1$ -> $C_2$) and control (baseline -> $C_1$) conditions were never present during learning. This means that if participants were able to extract rules about different item types, both transitions are matched in that they would be considered expectation violations. For example, baseline objects are only followed by A or other baseline items during learning, but in this task, they are followed by $C_1$ objects. Similarly, $C_1$ and $C_2$ objects are only preceded by B objects during learning, but in this task, $C_2$ objects are preceded by $C_1$ objects. Because neither set of transitions appears during learning, any difference in RTs across the conditions can be attributed to differences in the sequence-specific associations between objects, rather than rule-based information about sequential positions.

We additionally arranged B objects to always follow the A object from the same learning sequence. Like with the C objects, responses times for A and B were only analyzed if both objects were correctly endorsed as 'old' and if the prior object was also correctly endorsed. Note however that these comparisons were secondary to our main planned analyses, and due to constraints in randomization, A objects could either be followed by a different old object or a novel foil; thus the motor history for these objects is not controlled for to the same extent as C objects.

## Explicit memory task

This task followed the recognition priming task. Due to experimental error, two participants did not complete this task, leaving a sample size of 71. In this task, participants were first made aware that the objects in the learning task were arranged into sequences: 'You may have noticed that during the size task, some objects often appeared in sequences. For instance, maybe two objects always appeared back-to-back, Or, maybe two objects were always preceded by the same sequence.' This was accompanied by a visualization of a pair of overlapping $ABC_1$ and $ABC_2$ sequences. Participants then viewed pairs of objects and were asked to rate their familiarity from 'Not familiar' to 'Very familiar' specifically based on any sequence information they had learned. They responded by clicking on an un-numbered sliding scale. Each pair was presented for 6 s, separated by a jittered ITI ranging from 0.5 to 1.5 s. Participants viewed 60 pairs: 12 intact AB pairs; 12 re-arranged A and B objects; 12 intact C pairs; 12 re-arranged C objects; and 12 pairs of randomly paired baseline objects. All objects from sequences were viewed twice, once with its corresponding pair and once as a foil in a re-arranged pair.

## Unreported tasks

Participants from Experiment 1 were originally divided into two separate behavioral cohorts that piloted the feasibility of other memory tests that were ultimately not included in Experiment 2. One cohort (*N* = 48) completed a free recall task, and a second cohort (*N* = 42) completed a sequence test that mirrored the learning procedure but intermixed sequences from both learning sessions. These tasks were conducted at the end of the second session. Since all aspects of the procedure were

identical across the two cohorts until the presentation of those tasks, we have analyzed these two cohorts together to serve as Experiment 1.

## Statistical analyses

Linear mixed-effects models were used to quantify recognition priming and to characterize changes in RTs and accuracy across learning. Participant intercepts and slope terms for each included predictor variable were modeled as random effects. Random effects were kept maximal except where needed to avoid singular model fits (*Barr et al., 2013*). The significance of a given contrast was obtained using Satterthwaite approximate degrees of freedom, resulting in *F* or *t* statistics and corresponding p values. RTs for all learning trials were included in these models, but RTs during recognition priming were only included if the current and prior objects were correctly endorsed as 'old' (see Recognition Priming Task for more details). Accuracy was computed for any learning trial in which the preceding object was coded to be a different size (e.g. an object bigger than a shoebox following an object smaller than a shoebox), as comparisons among objects that were both bigger or both smaller are likely more variable and dependent on participants' subjective opinion.

Explicit memory performance was aggregated for each participant as the average difference in familiarity between Intact and Re-arranged pairs, such that a value of 0 would indicate no reliable discrimination and 1 would indicate maximal discrimination between the two conditions. These values were entered into two-tailed paired *t*-tests to test for differences in behavioral integration between the two learning sessions. One-sample *t*-tests with $\mu = 0$ were used to test for reliable above-chance performance, which would reflect evidence for integration.

## Experiment 2

### Subjects

Twenty-eight right-handed, native English speakers (14 female, mean age: 27.14, range: 20–34) participated in this experiment. Participants were recruited from New York University and the broader community. The University Committee on Activities Involving Human Subjects approved all recruitment and consent protocols. Three participants were excluded due to scanner malfunction, and one withdrew after the first session, leaving 24 participants that were included in the following analyses.

### Procedure

The experiment consisted of two sessions, separated by approximately 24 hr. The encoding and priming tasks were identical to that of Experiment 1, except that encoding was split into eight scans rather than four blocks. Additionally, in the first session, participants also first completed a resting-state scan, and then viewed all object stimuli in a random order (pre-learning exposure). Then, participants incidentally encoded sequences of the objects while performing a cover task (remote learning). Finally, participants completed a second resting-state scan. The stimuli were projected onto a screen in the bore of the scanner, and participants viewed them through a mirror attached to the head coil.

In the second session, participants began by incidentally learning a new set of object sequences while performing the same cover task as in the first session (recent learning). After a resting-state scan, participants completed several tasks consisting of both sets of stimuli from the two learning sessions. First, they completed a priming task, and again viewed all stimuli in a random order (post-learning exposure). Finally, participants completed an explicit test of memory integration.

The recognition priming task was the only task in the experiment that was not scanned, even though participants completed the task in the scanner between the rest and post-learning similarity scans.

### Explicit memory task

At the end of the recent session, participants underwent two scans that tested explicit memory for associations between A and B objects and associations between $C_1$ and $C_2$ objects (*Figure 2B*). The same instructions as Experiment 1 were used to make participants aware of the object sequences they had viewed. Then, in the first scan, participants were presented with an A object at the top of the screen and two B objects at the bottom of the screen. Participants were instructed to choose which B object was paired with the cued A object during learning. The foil B object was from a different

sequence in the same learning session. These trials were intermixed with trials where B objects were cued and participants chose the corresponding A object. The same cue/foil structure was used to test explicit memory for the association between $C_1$ and $C_2$ objects in the second scan. Each trial was presented for 4 s with a variable ITI ranging from 3 to 8 s.

### Pre- and post-learning snapshots

Before the remote learning session on Day 1 and after the recent learning session and priming task on Day 2, participants viewed all objects in a pseudo-random order over the course of two 11.8-min scans. The goal of these scans was to create a template pattern for each object stimulus before and after participants learned their temporal associations. With these template patterns, we measured changes in representational similarity between objects that were associated through their temporal structure. Changes in pattern similarity from pre- to post-learning among *recently* learned objects would reflect changes in memory representations that corresponded to the temporal structure learned a few minutes beforehand. In contrast, changes in pattern similarity among *remotely* learned objects would reflect consolidation-related processes in addition to learning-related changes seen in the recent condition.

Each object was presented for 2 s, separated by a variable ITI ranging from 2 to 5 s. Participants were instructed to view each object and press a button if a small pound sign (#) appeared anywhere on the image. A pound sign appeared on 20 images (approximately 8% of all trials presented). The objects for which a pound sign appeared were presented a total of five times instead of four, and the trial with the pound sign was omitted from all similarity analyses.

All object stimuli were presented across two scans each in the pre- and post-learning exposure phases. The order during the pre-learning exposure was identical to the order during the post-learning exposure in order to equate any potential confounds due to order effects or biases in modeling the BOLD response. The order of the objects in each scan was randomized for each participant, with the constraint that no object would repeat back-to-back. All A objects, all $C_1$ objects, and half of the baseline objects were presented four times, intermixed in the first scan, and all B objects, all $C_2$ objects, and the other half of the baseline objects were presented four times, intermixed in the second scan. This arrangement enabled us to prioritize the analysis of similarity between A and B objects, and $C_1$ and $C_2$ objects, since the objects were presented in separate scans (*Mumford et al., 2014*). See *Table 1C and D* for transition probabilities of objects in the two scans.

### Resting-state scans

Participants completed three 6-min resting-state scans throughout the experiment: the first and last scans of the remote session, and immediately after sequence learning during the recent session (pre-learning, post-remote learning, and post-recent learning). During the scans, participants viewed a blank gray screen and were instructed to remain awake while thinking about whatever they wanted (*Greicius et al., 2003*; *Tambini et al., 2010*).

### fMRI parameters

All scans were collected with a whole-head coil using a 3T Siemens Allegra MRI system. Functional scans consisted of multi-echo gradient-echo planar images (EPI: 2000 ms repetition time (TR), 15 ms echo time (TE), flip angle = 82°, field of view (FOV) = 192 × 240, 3 mm isotropic voxels), with 34 slices oriented parallel to the anterior commissure - posterior commissure (AC–PC) line. For both sessions, a customized calibration scan was collected using the same slice prescription as the EPI scans. At the end of the second scan, a T1-weighted high-resolution magnetization-prepared rapid-acquisition gradient echo (MPRAGE) sequence (1 × 1 × 1 mm voxels, 176 sagittal slices) was collected.

### Preprocessing

All learning, rest, and pre- and post-learning exposure scans underwent the same preprocessing steps with FSL (FEAT: http://www.fmrib.ox.ac.uk/fsl). The first six volumes of each EPI were discarded to allow for scanner stabilization. Then, each scan was slice-time corrected and realigned to correct for motion within each run. Smoothing differed based on the analysis: the learning and rest scans were smoothed with a 6-mm full width at half maximum (FWHM) Gaussian kernel, while the pre- and

post-learning scans were smoothed with a 3-mm FWHM Gaussian kernel. All scans were high-pass filtered at 0.01 Hz to remove low-frequency drifts in signal (*Cordes et al., 2001*) and then aligned to the first pre-learning similarity scan from the remote session.

## Pre- and post-learning pattern similarity

The four pre- and post-learning scans were entered into separate general linear models (GLMs) after preprocessing. In each scan, a separate 3-s boxcar regressor was created for the four presentations of each object throughout the scan. One 'junk' regressor was created to model the onset of all trials for which participants made a key press. These trials could either be the target trials presented with the pound sign, or any other trial where the participant incorrectly pressed the button in response to a stimulus with no pound sign. As with the learning scans, regressors accounting for head motion were included as well.

These GLMs gave rise to two $t$-statistic maps for each object presented in the two learning sessions, one from before learning and one from after learning. For each map, the spatial pattern of activity across each ROI was extracted into a vector. These vectors were not $z$-scored; similarity analyses did not meaningfully change if voxels were $z$-scored across all patterns. Pearson correlations were computed to measure the neural similarity between vectors of all C objects presented. These were organized into two separate matrices representing C objects from the recent and remote learning sessions. The matrices were then transformed into vectors with the same order of matrix elements. These two vectors were then correlated with the neural 'integration' model, a matrix of ones and zeros that represented whether C objects were from overlapping or distinct sequences, that was similarly organized into vector form (e.g. *Figure 3A*). A third vector was computed by subtracting the pre-learning vector from the post-learning vector to represent learning-related changes in neural similarity (e.g. *Figure 3B*). A high Point-biserial correlation between neural data and this integration model would reflect greater similarity between images that followed the same AB pairs relative to images that followed different AB pairs. These correlations were Fisher transformed before being entered into statistical tests.

To control for the possibility that similarity between two objects was driven by differences in average BOLD signal between them, we generated two additional correlation matrices composed of $C_1$ and $C_2$ objects, arranged in the same order as the pattern similarity and integration matrices, separately for the pre- and post-learning exposure scans. For each cell, we computed the absolute value of the difference in BOLD signal across the two objects. This matrix thus represents cases where BOLD signal of two objects is close or far apart. Then, since the relationship between the neural integration model and similarity data for a given ROI and exposure phase was originally analyzed by transforming the two matrices into vectors and computing their correlation, we also transformed the BOLD signal matrix into a vector with the same order of elements and used it in partial correlation analyses that corresponded to the main analyses (e.g. re-computing the correlation between mPFC similarity and the neural integration model, while accounting for the difference in BOLD signal).

## Resting-state connectivity

The rest scans were used to quantify low-frequency correlations between pairs of ROIs (*Albert et al., 2009*; *Tambini et al., 2010*; *Tompary et al., 2015*), a measure of functional connectivity between regions. To this end, the preprocessed data from the three rest scans were band-pass filtered leaving signal between 0.01 and.1 Hz (*Fox et al., 2005*). They were then entered into separate GLMs to model nuisance signals, including: six motion regressors and their temporal derivatives; stick functions accounting for sudden head movements; and nuisance signals from white matter tissue and cerebral spinal fluid (CSF) and their temporal derivatives. The motion regressors were derived from frame displacement measurements identifying during motion correction. TRs with sudden head movements were identified with *FSL_motion_outliers*. To create nuisance regressors, each participant's MPRAGE was segmented into separate masks comprising gray matter, white matter, and CSF, using *FAST*. The gray matter and CSF masks were aligned to each participant's functional volumes and then eroded using *fslmaths*, to minimize the likelihood that these masks contained voxels that partially overlapped with gray matter. Then, the average time course across all voxels in each mask was extracted from the preprocessed rest scans. These time courses were entered into each run's GLM along with their temporal derivatives.

The residuals of these GLMs were band-pass-filtered, leaving signal ranging from 0.01 and 0.1 Hz, which is the frequency range known to correspond to correlations between gray matter regions in functional neuroimaging data (*Cordes et al., 2001*). Then, the average time course for every volume in each rest run was extracted for each ROI. These time courses were then correlated (Pearson correlation), Fisher transformed, and entered into statistical tests.

### Regions of interest

Bilateral hippocampus was defined using FSL's automatic subcortical segmentation protocol (FIRST), which anatomically defines subcortical regions using each participant's T1 anatomical image. To isolate anterior and posterior portions, the number of slices in each hemisphere was divided into three sections, and the anterior and posterior thirds were used as ROIs. All ROIs were resampled and aligned to the first similarity scan during the remote session, consistent with the functional scans.

Medial PFC and LOC were functionally defined from the learning scans. Specifically, we sought regions that were sensitive to the transition probabilities of the presented triplets. To identify changes in BOLD signal to objects with different transition probabilities, we entered each learning scan into a separate GLM after preprocessing. Two regressors were generated: predictable (B, $C_1$, and $C_2$), and unpredictable (A, baseline). $C_1$ and $C_2$ objects were included in the same regressor because participants were unable to differentiate those objects during learning. Trials with no response, or with an RT >3 SD from the participant's mean RT, were separately modeled in a regressor of no interest. These regressors and their temporal derivatives were convolved with FSL's canonical hemodynamic response

**Table 2.** R and L indicate right and left hemispheres.

Extent is size of the clusters in voxels. x, y, and z coordinates indicate the center of gravity in MNI space (mm). p corresponds to corrected significance of cluster. * indicates clusters used for ROIs.

| Region | Extent | x | y | z | p |
|---|---|---|---|---|---|
| *Predictable > unpredictable* | | | | | |
| *R Parahippocampal, lateral occipital cortex | 4644 | 28.1 | −77.8 | −3.82 | <0.001 |
| *L Parahippocampal, lateral occipital cortex | 2623 | −29.1 | −80.3 | −5.08 | <0.001 |
| L Angular gyrus, central opercular cortex | 1246 | −52.1 | −43.3 | 27.1 | <0.001 |
| * Medial prefrontal cortex, frontal pole | 1195 | −2.43 | 54.5 | 4.31 | <0.001 |
| R Angular gyrus, middle temporal gyrus | 911 | 62.2 | −41.2 | 10.1 | <0.001 |
| Posterior medial cortex | 675 | −0.133 | −47.5 | 34 | <0.001 |
| Middle temporal gyrus, posterior | 372 | −56.2 | −24.9 | −5.67 | <0.001 |
| Posterior cingulate gyrus | 308 | 1.43 | −21 | 39.6 | <0.001 |
| R Parietal operculum cortex | 217 | 59.7 | −24.8 | 21.1 | <0.001 |
| R Inferior frontal gyrus | 197 | 55.2 | 31.3 | −1.84 | 0.001 |
| L Putamen | 168 | −28.5 | −10.8 | 5.01 | 0.003 |
| L Middle temporal gyrus, anterior | 142 | −51.6 | −1.3 | −23.9 | 0.008 |
| R Frontal Pole | 98 | 16 | 53.2 | 28.1 | 0.04 |
| *Unpredictable > predictable* | | | | | |
| L Insula, inferior and middle frontal gyrus | 1788 | −39.7 | 20.1 | 12 | <0.001 |
| R Insula, inferior frontal gyrus | 802 | 36.4 | 19.8 | 4.41 | <0.001 |
| L Supramarginal gyrus | 252 | −39.4 | −43.7 | 43.1 | <0.001 |
| Paracingulate cortex | 221 | −0.922 | 25.8 | 36.4 | <0.001 |
| R Middle frontal gyrus | 199 | 44.3 | 33.7 | 21.4 | 0.001 |
| L Inferior temporal gyrus | 142 | −52.7 | −55.2 | −10.6 | 0.008 |
| L Superior lateral occipital cortex | 98 | −22.6 | −65.7 | 38.5 | 0.04 |

function (HRF). To account for head motion, the six regressors derived from the motion correction procedure were included in each GLM along with their temporal derivatives and stick function regressors derived by *FSL_motion_outliers*.

This resulted in 16 models per participant (8 per session), which were then entered into a fixed-effects analysis in which the first-level estimates were averaged over the two sessions. The resulting contrasts revealed clusters of voxels whose BOLD signal was reliably different for predictable versus unpredictable objects. These estimates were averaged together at the group level in a random-effects analysis. Clusters were determined using a statistical threshold of $z > 3.1$ and a corrected cluster significance threshold of $p < 0.05$ using FSL's Threshold-Free Cluster Enhancement.

This analysis revealed several regions including mPFC and LOC (*Table 2*). As the clusters that overlapped with our target ROIs encompassed other anatomically distinct areas, we masked both ROIs to constrain their coverage. To create the mPFC ROI, we masked the cluster with areas A14m and A10m (*Audrain and McAndrews, 2022*) from the Brainnetome atlas (https://atlas.brainnetome.org/). To create a bilateral LOC ROI, we masked the two clusters extending over right and left lateral occipital cortex and parahippocampal cortices with the top 90% voxels of the 'Lateral Occipital Cortex, Inferior Division' in the Harvard-Oxford Cortical probabilistic atlas.

## Statistical tests

Analysis of the behavioral variables was identical to the procedure form Experiment 1, except that explicit memory was aggregated for each participant as the proportion of responses that corresponding to the C object with the same overlapping sequence as the target C object. For the neural data, repeated-measures analyses of variance and two-tailed paired *t*-tests were used to test the significance of group-level effects. Pearson correlations were used to quantify relationships between neural and behavioral measures across participants. To examine how rest connectivity and mPFC similarity related to recognition priming (*Figure 5B*), a multiple linear regression was computed with both neural measures as predictors, and the significance of these was calculated with *F* values and corresponding p values. Their unique variance explained was determined with semi-partial $R^2$ values.

Results were Bonferroni-corrected for multiple comparisons, depending on the number of regions or region pairs for which a particular analysis was computed. Pattern similarity across C objects was corrected for four regions (mPFC, LOC, anterior hippocampus, posterior hippocampus). Pattern similarity across A and B objects was corrected for three regions (LOC, anterior hippocampus, posterior hippocampus). Learning-related changes in rest was corrected for five region pairs (posterior hippocampus—LOC, anterior hippocampus—LOC, mPFC—LOC, posterior hippocampus—mPFC, anterior hippocampus—mPFC).

## Acknowledgements

Thank you to Max Bluestone and Yuri Jiao for help with data collection, Emily Cowan for helpful discussions and comments on the manuscript, and Sarah Dubrow for encouragement and mentorship in early stages of the project. The study was supported by National Institute of Mental Health Grant MH074692 (LD), Dart Neuroscience (LD), and NSF Graduate Research Fellowship Program (AT).

## Additional information

### Competing interests

Lila Davachi: Reviewing editor, *eLife*. The other author declares that no competing interests exist.

### Funding

| Funder | Grant reference number | Author |
| --- | --- | --- |
| National Institute of Mental Health | MH074692 | Lila Davachi |
| Dart Neuroscience | | Lila Davachi |

| Funder | Grant reference number | Author |
|---|---|---|
| National Science Foundation | GRFP | Alexa Tompary |

The funders had no role in study design, data collection, and interpretation, or the decision to submit the work for publication.

## Author contributions

Alexa Tompary, Conceptualization, Data curation, Software, Formal analysis, Validation, Investigation, Visualization, Methodology, Writing - original draft, Project administration, Writing - review and editing; Lila Davachi, Conceptualization, Supervision, Funding acquisition, Methodology, Project administration, Writing - review and editing

## Author ORCIDs

Alexa Tompary https://orcid.org/0000-0001-7735-3849
Lila Davachi https://orcid.org/0000-0003-4317-0889

## Ethics

Consent to participate and consent to publish was obtained for all participants prior to beginning the first experimental session. Open their arrival to the session, participants read a consent document and signed at the bottom. New York University's University Committee on Activities Involving Human Subjects approved all recruitment and consent protocols (HS#10-0090).

## Decision letter and Author response

Decision letter https://doi.org/10.7554/eLife.84359.sa1
Author response https://doi.org/10.7554/eLife.84359.sa2

---

# Additional files

## Supplementary files

• MDAR checklist

## Data availability

Source data files have been provided for all figures, including figure supplements. Raw data and analysis code required to reproduce figures and statistics is provided for the behavioral data from both experiments. Similarity and connectivity values are provided with analysis code needed to reproduce all figures and reported statistics. All of these materials are currently publicly available on Open Science Framework. Unprocessed brain data is available on OpenNeuro.

The following datasets were generated:

| Author(s) | Year | Dataset title | Dataset URL | Database and Identifier |
|---|---|---|---|---|
| Tompary A | 2023 | Integration of overlapping sequences emerges with consolidation through mPFC neural ensembles and hippocampal-cortical connectivity | https://osf.io/es8zj | Open Science Framework, es8zj |
| Tompary A, Davachi L | 2024 | Integration of overlapping sequences emerges with consolidation through mPFC neural ensembles and hippocampal-cortical connectivity | https://doi.org/10.18112/openneuro.ds005581.v1.0.0 | OpenNeuro, 10.18112/openneuro.ds005581.v1.0.0 |

---

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
