## [Editor Report]

This important study investigates how memory representations are transformed over time (24h period), using a novel behavioral task and fMRI. The work advances our understanding of the neural processes supporting the behavioral integration of memories for distinct events that are never experienced together in time but are linked by shared predictive cues. Evidence supporting the claims is convincing, with inclusion of important control analyses that lend support to the authors' interpretation of the data.

---

## [Decision Letter]

**Decision letter after peer review:**

Thank you for submitting your article "Consolidation-dependent behavioral integration of sequences related to mPFC neural overlap and hippocampal-cortical connectivity" for consideration by *eLife*. Your article has been reviewed by 3 peer reviewers, one of whom is a member of our Board of Reviewing Editors, and the evaluation has been overseen by Michael Frank as the Senior Editor. The reviewers have opted to remain anonymous.

Essential Revisions:

(1) Reviewers agreed that more data on the neural integration/differentiation effects are key to evaluating the findings (points originally raised by Reviewer 1, comment 2 [R1.2] and R3). Please report the pattern underlying the overall pre- to post-learning change in correspondence with the predicted model: how similar are related and unrelated C items to one another pre-learning? How similar are they post-learning? At both pre- and post-learning, is there a difference between related and unrelated C similarities? Is there a reliable pre- to post-learning change in one or both of these sets of comparisons? Once these analyses are fully reported, please revise the paper if needed such that the interpretation is consistent with these more specific patterns of change. Moreover, please consider which effects or changes seem to be driving the subsequent analyses that use these measures.

(2) Please consider in the paper how the lack of a same-day baseline for the recent condition would affect the connectivity changes. Reviewers wondered why there was no day 2 baseline, and whether this was for theoretical or practical reasons [Reviewer 1.1a-b]. In the consultation session, Reviewers expressed different intuitions about whether indexing pre- to post-learning change for the recent memories using a baseline from a prior day should lead to inflated, or noisier (R1.1c-d), estimates for that measure. However, there was agreement that this issue should be discussed as a limitation in the paper. In addition, Reviewers thought that perhaps control analyses with some set of regions that should be minimally involved in memory consolidation might help strengthen the current findings.

(3) Many of the effects were only observed in the remote condition or for remote items, but not for the recent (R3). However, there were no significant interactions such that the effects are reliably larger for remote than recent. Therefore, the interpretive contrast rests on a null effect in the recent condition, which of course could be due to many factors (including things like the lack of same-day baseline, as noted in the previous point and mentioned by R1; as well as perhaps insensitivity to behavioral priming for the recent memories due to speeding in the control condition as raised by R3). Accordingly, Reviewers all agreed that claims should be toned down to align with these statistics.

(4) There were a few points raised about the different sequences participants were exposed to across the entire study. (a) First, Reviewers wanted to see more information about this. For example, R2 (weakness #1) recommended a table of transition probabilities during sequence learning and the priming task, and consideration of whether those do or do not match. It might additionally be useful to include this information for the pre- and post-learning exposure phases. Could correspondence, or lack thereof, of the different transitions across tasks have influenced the results in any way? All Reviewers agreed that more information on that structure would be valuable. (b) Given the priming occurred prior to the post-learning exposure, it seemed as though the priming task could influence the post-learning representations. Specifically, in the Reviewers' understanding, unrelated C items always appeared in sequence during the priming task and therefore could have become "integrated" through that experience (R1.3). Please clarify this issue, perform control analyses, and/or consider this limitation in the Discussion.

(5) Reviewers had questions about the priming task. First, there were requests for more details about this task design in the main paper. Second, Reviewers noted that priming was mainly driven by slowing for the control items (with remote RTs being longer than recent RTs), rather than a difference for the critical C items (R3). Third and perhaps relatedly, Reviewers thought that control transitions always happened prior to the primed transitions (R1.4), and wondered if this might somehow be important (i.e., could it explain the control item slowing somehow, or more generally have influenced the findings?). For the latter two subpoints, Reviewers wanted greater consideration of these issues in a revision -- perhaps analytically if possible, and if not then noted in the paper as a limitation. Overall however, the Reviewers reasoned that the design in which recent and remote memories were tested together, as well as the specificity of the effects of interest to the remote condition, seemed to make some of the "less interesting" explanations like general task practice less likely.

(6) It seems that given the priming task design and as noted in #4 above, C items that were indirectly related always appeared together in the priming task, and never appeared with unrelated C items. Given this, the Reviewers thought that the behavioral results could not distinguish between the integration of the related C items in particular (i.e., those that occurred at the end of the same sequence), as opposed to a more general integration of all C items with one another (i.e., perhaps all end-of-sequence items become more similar; point raised originally by R3). Can the authors speak to this concern? If the authors can perform some control analysis to address this issue, that would be ideal. If not, then acknowledgment of a limitation about lack of behavioral specificity for the related C's (in particular, vs. the C items overall) becoming integrated or differentiated would be warranted.

(7) Please be cautious in phrasing with respect to the connectivity changes as a result of learning. In particular be sure they are not being discussed as "reactivation," which the Reviewers all agreed was not measured directly (a point raised originally by R3).

(8) In the Introduction, two Reviewers (R1 and R2) thought it would be worth better motivating the regions of interest (why posterior but not anterior hippocampus; and why LOC and not mPFC?). Please consider these suggestions in revising your Introduction.

(9) Reviewers would like to see a sharper distinction between planned and exploratory analysis, and multiple comparisons correction for exploratory (R2).

(10) If the authors see fit, they may want to comment on the implicit/explicit distinction. In particular, it would be helpful to know whether they think the explicit elements in their task (e.g., like recognition memory judgments in the priming task) are crucial to the findings (R2), in light of how systems consolidation models are generally conceptualized as pertaining more to explicit memory (R3).

(11) Reviewers had numerous other suggestions and questions, which I would ask the authors to additionally consider.

*Reviewer #1 (Recommendations for the authors):*

(1) Did the posterior hippocampus show a change in neural integration for either recent or remote (as shown for the anterior in Supplemental Figure 6)? Would be nice to report this for completeness.

(2) p. 6 – here they focus on posterior hippocampus connectivity with category-sensitive regions of the neocortex, which I think is fine. They note also that this choice is exploratory. But, they in their motivation do not really touch on the fact that integration itself has generally been more ascribed to the anterior hippocampus-so it seems to me that there is a possibility that either region could be important, and if the authors want they could perhaps set those dual possibilities up. Perhaps replay via the posterior hippocampus of the single events with later integration in the neocortex is one mechanism, but online integration & replay of those associations (perhaps more of an anterior hippocampus operation) is another. It then perhaps becomes clearer why it is interesting that the authors find the effect specific to the posterior hippocampus, with nothing much going on in the anterior. Could the work speak to these two possibilities (i.e., the findings seem more consistent with the first possibility of immediate storage for the memories separately, and offline integration via reactivation of the constituent traces)? Alternatively, might the authors explain why they think integration per se in the anterior hippocampus seen in previous work should not be related to the post-encoding connectivity profile?

(3) p. 7, line 142 – It was not clear to me why RT (per se, or in all cases) is an ideal readout of memory integration. Can the authors provide more explanation on this point?

(4) It was unclear from the Introduction what the reason was for the separate behavioral experiment. Up until p. 13, it seems as though Experiment 2 basically serves to replicate & extend Experiment 1, and only later is it clear that there are interesting nuanced differences in the explicit test. Can this be stated earlier (in the Intro or earlier in the Results)?

(5) p. 9 – For the behavioral results (RT) for the sequence learning task, please clarify whether only correct responses were included in this model. Or, if all trials, it would be helpful to clarify why the decision to include incorrect trials as well.

(6) It took me a while to work out exactly how the priming task was set up, and I think more details in the main paper would have been useful. It was unclear to me until I read the Methods that every sequence had one C item that appeared in the "primed" condition and one in the "control" condition (and that, as noted above, the control transition always has to happen first, by definition). I think it would be helpful to be more specific in the main paper about the critical comparison-specifically, more on the control condition. On first read, I was not sure whether C trials preceded by novel foils were used as a control (which would not be ideal, but I don't think this is what the authors did). I think baseline objects were old objects and were presented an equal number of times as the C items (based on p. 35); putting this in the main text would have been so helpful. It might also be helpful to note that each object appeared just once in the priming task, to help readers get a sense of how many trials/sequences (or the max #) go into each comparison.

(7) I was a bit confused about the use of the words "baseline" and "control" objects (e.g., on p. 11)-it seemed at times the terms were being used interchangeably, and in other places, the baseline objects were what made up the control condition.

(8) p. 13 – at the end of the section on priming, possibly worth qualifying the final sentence to note the effect of order as being present (interestingly) only for the remote sequences?

(9) p. 13, line 317 – where it says accuracy was reliably above chance, I think that means there is significant evidence for integration (more familiarity for intact vs. rearranged pairs), but I wasn't totally sure since it wasn't called accuracy earlier. Please clarify.

(10) For p. 15 – "all object images" presented during the pre- and post-learning exposure phases: I assume this is all objects from the remote and recent learning tasks, not the novel foils from the priming task.

(11) p. 17 – I was confused about this partial correlation control analysis. Did this analysis use a similar approach to the main analysis, where the authors took some kind of average score per participant? (Would that be average across the matrix diagonal, i.e., between all pairs of related C objects?) Or, was this somehow used to control for differences in BOLD signal between C1 and C2 in a trial-by-trial way?

(12) p. 38 – "Accuracy computed for any learning trial in which the preceding object was coded to be a different size (e.g. an object bigger than a shoebox following an object smaller than a shoebox), as comparisons amongst objects that were both bigger or both smaller are likely more variable and dependent on participants' subjective opinion." line 987-990 – is there a portion (clause? verb?) of this sentence missing?

(13) p. 39 – what instructions were given prior to the explicit memory test, especially on how to do the C1-C2 association trials?

(14) I thought it was interesting that the authors decided to match the C1 and C2-related objects in terms of size judgment so that the motor history was the same. I was curious as to whether the authors thought this might have encouraged integration since both objects point toward the same response (vs. a case where differentiating the two objects might be more important, as when the response depends on object identity). Just a thought!

(15) The authors were generally very thorough in their references to the literature, but they might want to consider citing these additional papers that are relevant: Abolghasem et al. (2023) Developmental Science; Ye et al. (2020) *eLife*. Also, since the Audrain and colleagues' paper is now published, please be sure to switch the citation from the biorxiv.

*Reviewer #2 (Recommendations for the authors):*

(1) A table outlining transition probabilities between all relevant item types in the sequence learning and the implicit recognition priming task would be helpful.

(2) Also relating to the public review, a clear line should be drawn between planned and exploratory analyses, and multiple comparison corrections should be applied to exploratory analyses that perform the same contrast on multiple ROIs.

(3) Statistics should be reported for the analysis of z-scoring across voxels to control for the influence of the overall BOLD signal on the neural pattern analysis.

(4) For the partial correlation control analysis, the methods and results should be clear about whether the BOLD differences between C1 and C2 represent pre- or post-learning distances (or both)?

(5) Work by the Kuhl lab on the repulsion of overlapping memories (including spatial sequence memories) appears highly relevant and should be discussed.

(6) The sequence learning (i.e., size judgment) task puts a strong emphasis on the temporal relationships between items, and the priming task has a strong explicit recognition memory component. Could the authors briefly comment on whether they believe their results generalize to entirely implicit settings with no such explicit temporal binding or explicit memory demands?

*Reviewer #3 (Recommendations for the authors):*

(1) The validity of the observed behavioral priming effect could be more rigorously tested to improve the interpretation. Currently, the authors compare the reaction times of target C2 preceded by C1 from the overlapping sequence (prime condition), with a condition where C1 is preceded by a baseline object that is not a part of any sequence (control condition). Do they still see a priming effect when C1 is preceded by a C object from a non-overlapping sequence instead of the baseline objects (is this even possible with the task design)? This would test the specificity of the priming effect. If C objects are integrated due to the overlapping components of their sequences, one should observe priming in this condition but not when preceded by C objects from non-overlapping conditions. Alternatively, C objects could all be integrated with each other due to some other dimension of the task, such as the fact that C objects are presented less frequently than A or B or are less predictable in general than B, etc.

(2) It would be helpful to get a sense of the overall model fits for the neural data to the integration model in the mPFC and LOC, as we only see the change in fit, making it difficult to understand how well these models fit the data in the first place. In addition, could a decrease in fit be due to integration occurring across C items from non-overlapping sequences, instead of differentiation of C items from overlapping sequences?

(3) For the neural measures related to the priming section (page 23), I was confused as to why the authors took a model comparison approach to determine the unique variance explained by each predictor. Isn't it the case that when you include both similarity and connectivity in the model, if both remain significant predictors, they explain unique variance in priming? You could then calculate the semi-partial R2 to determine exactly how much unique variance is explained by each. I point this out because model comparisons are anti-conservative given that a model with more predictors will inherently explain more variance than a model with fewer predictors.

(4) The authors frequently refer to their connectivity results as reflecting post-encoding reactivation. I understand that there is evidence from other studies that reactivation occurs during the post-encoding period, but the authors have not shown this (I believe one would need to see patterns evident of encoding being reinstated during the post-encoding period). It would be more accurate to refer to the results in terms of post-encoding coupling rather than reactivation throughout, so as not to over-interpret in light of the data.

---

## [Author Response]

Essential Revisions:(1) Reviewers agreed that more data on the neural integration/differentiation effects are key to evaluating the findings (points originally raised by Reviewer 1, comment 2 [R1.2] and R3). Please report the pattern underlying the overall pre- to post-learning change in correspondence with the predicted model: how similar are related and unrelated C items to one another pre-learning? How similar are they post-learning? At both pre- and post-learning, is there a difference between related and unrelated C similarities? Is there a reliable pre- to post-learning change in one or both of these sets of comparisons? Once these analyses are fully reported, please revise the paper if needed such that the interpretation is consistent with these more specific patterns of change. Moreover, please consider which effects or changes seem to be driving the subsequent analyses that use these measures.

We have included several new figures and statistics that target these questions and we are pleased with the additional evidence that this provides for our claims. Specifically, we included visualizations of model fits separately for pre- and post-learning exposure scans. We also have included visualizations of the average similarity for related and unrelated C pairs (and AB pairs as well). In brief, all changes in model fits are driven by changes in similarity between the related pairs rather than the unrelated pairs, which provides strong evidence that our integration and differentiation effects were driven by learning and consolidation of the object sequences. Please see our response to R1.2 for more detail and the included visualizations.

(2) Please consider in the paper how the lack of a same-day baseline for the recent condition would affect the connectivity changes. Reviewers wondered why there was no day 2 baseline, and whether this was for theoretical or practical reasons [Reviewer 1.1a-b]. In the consultation session, Reviewers expressed different intuitions about whether indexing pre- to post-learning change for the recent memories using a baseline from a prior day should lead to inflated, or noisier (R1.1c-d), estimates for that measure. However, there was agreement that this issue should be discussed as a limitation in the paper. In addition, Reviewers thought that perhaps control analyses with some set of regions that should be minimally involved in memory consolidation might help strengthen the current findings.

We agree that this is a major limitation of the manuscript. As a summary of the manuscript revisions, we added two new analyses of hippocampal-LOC coupling, one correlating these measures with response priming separately for pre-and post-learning scans, and one confirming no difference in variance in the change scores between the two learning sessions. Please see the point-by-point responses below for the specific findings and our responses to reviewer’s specific points about this.

Furthermore, our thinking about the lack of a same-day baseline for the recent condition spurred additional considerations of our design, namely the fact that we split encoding across two sessions rather than combing encoding into one session and splitting retrieval across an immediate and delayed session. We fleshed out the Discussion section for both of these points.

(3) Many of the effects were only observed in the remote condition or for remote items, but not for the recent (R3). However, there were no significant interactions such that the effects are reliably larger for remote than recent. Therefore, the interpretive contrast rests on a null effect in the recent condition, which of course could be due to many factors (including things like the lack of same-day baseline, as noted in the previous point and mentioned by R1; as well as perhaps insensitivity to behavioral priming for the recent memories due to speeding in the control condition as raised by R3). Accordingly, Reviewers all agreed that claims should be toned down to align with these statistics.

We have rephrased our interpretation of these findings and more explicitly note the lack of significant interactions. Please see our responses to R1 and R3 for more detail.

(4) There were a few points raised about the different sequences participants were exposed to across the entire study. (a) First, Reviewers wanted to see more information about this. For example, R2 (weakness #1) recommended a table of transition probabilities during sequence learning and the priming task, and consideration of whether those do or do not match. It might additionally be useful to include this information for the pre- and post-learning exposure phases. Could correspondence, or lack thereof, of the different transitions across tasks have influenced the results in any way? All Reviewers agreed that more information on that structure would be valuable. (b) Given the priming occurred prior to the post-learning exposure, it seemed as though the priming task could influence the post-learning representations. Specifically, in the Reviewers' understanding, unrelated C items always appeared in sequence during the priming task and therefore could have become "integrated" through that experience (R1.3). Please clarify this issue, perform control analyses, and/or consider this limitation in the Discussion.

These details are crucial for readers to understand to be able to evaluate and interpret our findings, and we thank the reviewers for highlighting a need for their clarification. (a) We have included a table of transition probabilities and some discussion of how they differ across the learning and recognition priming tasks (see response to R2). (b) We have included some justification for why we arranged for the priming task to occur before the exposure scans; we agree that this is a potential caveat to the design of our study, but we are encouraged by the fact that the neural effects are specific to remotely learned sequences even though sequences from both conditions are presented during priming in this “integrated” order (see response to R2 and R1.3).

(5) Reviewers had questions about the priming task. First, there were requests for more details about this task design in the main paper. Second, Reviewers noted that priming was mainly driven by slowing for the control items (with remote RTs being longer than recent RTs), rather than a difference for the critical C items (R3). Third and perhaps relatedly, Reviewers thought that control transitions always happened prior to the primed transitions (R1.4), and wondered if this might somehow be important (i.e., could it explain the control item slowing somehow, or more generally have influenced the findings?). For the latter two subpoints, Reviewers wanted greater consideration of these issues in a revision -- perhaps analytically if possible, and if not then noted in the paper as a limitation. Overall however, the Reviewers reasoned that the design in which recent and remote memories were tested together, as well as the specificity of the effects of interest to the remote condition, seemed to make some of the "less interesting" explanations like general task practice less likely.

We agree that the details of the priming task merit more description in the paper and have made substantial revisions to the manuscript to address these questions. We have added more detail about the task when introducing the results, clarified that the control transitions do indeed occur before the primed conditions, and added an analysis that demonstrates that while responses do become faster over the course of the task, the difference in RT for primed relative to control condition for the remotely learned objects remains robust when accounting for this effect. Additionally, we have added caveats to the discussion about the design choices we made for this task and how they may impact the results. Please see responses to specific reviewer comments for more detail.

(6) It seems that given the priming task design and as noted in #4 above, C items that were indirectly related always appeared together in the priming task, and never appeared with unrelated C items. Given this, the Reviewers thought that the behavioral results could not distinguish between the integration of the related C items in particular (i.e., those that occurred at the end of the same sequence), as opposed to a more general integration of all C items with one another (i.e., perhaps all end-of-sequence items become more similar; point raised originally by R3). Can the authors speak to this concern? If the authors can perform some control analysis to address this issue, that would be ideal. If not, then acknowledgment of a limitation about lack of behavioral specificity for the related C's (in particular, vs. the C items overall) becoming integrated or differentiated would be warranted.

It is true that we did not include a control condition in which C objects from different sequences appeared together, and we now explain our reasoning for this, and the potential drawbacks to this choice, in a discussion paragraph. However, we emphasize that only mPFC similarity amongst C objects from the same sequences correlated with the extent of this priming effect across participants; similarity amongst C objects from different sequences did not scale with priming. This is an encouraging finding that we discovered through our revision process and we believe it gives strength to our claim that priming is reflective of sequence-level associations rather than general integration of all C objects based on their position in each sequence.

(7) Please be cautious in phrasing with respect to the connectivity changes as a result of learning. In particular be sure they are not being discussed as "reactivation," which the Reviewers all agreed was not measured directly (a point raised originally by R3).

We now use the terms ‘coupling’ or ‘hippocampal-cortical coupling’ to refer to the measure and reported findings. We did keep the language we used to introduce the measure as a way to capture post-learning processing that may reflect reactivation, but we agree that reactivation is not directly measured in this experiment.

(8) In the Introduction, two Reviewers (R1 and R2) thought it would be worth better motivating the regions of interest (why posterior but not anterior hippocampus; and why LOC and not mPFC?). Please consider these suggestions in revising your Introduction.

We have expanded our motivation for each region of interest in the introduction and have also provided some speculation about the potential differences in function across anterior and posterior hippocampus. This has substantially strengthened the paper’s treatment of the possible neural computations that underpin consolidation-dependent integration.

(9) Reviewers would like to see a sharper distinction between planned and exploratory analysis, and multiple comparisons correction for exploratory (R2).

We have made several edits to the manuscript to clarify this distinction – please see our comments to R2.3 for more detail.

(10) If the authors see fit, they may want to comment on the implicit/explicit distinction. In particular, it would be helpful to know whether they think the explicit elements in their task (e.g., like recognition memory judgments in the priming task) are crucial to the findings (R2), in light of how systems consolidation models are generally conceptualized as pertaining more to explicit memory (R3).

This a great suggestion, and we have added some commentary on this distinction in the Discussion (pages 31-32, lines 824-838). Please see our responses to R2.6 and R3.3 for more details.

(11) Reviewers had numerous other suggestions and questions, which I would ask the authors to additionally consider.

Please see below for our responses to each point.

Reviewer #1 (Recommendations for the authors):(1) Did the posterior hippocampus show a change in neural integration for either recent or remote (as shown for the anterior in Supplemental Figure 6)? Would be nice to report this for completeness.

This change in integration was selective to the anterior hippocampus; we report that it was not observed in posterior hippocampus – neither for A-B pairs (page 21, lines 558 – 560) nor C_1_-C_2_ pairs (page 21, lines 538-540).

(2) p. 6 – here they focus on posterior hippocampus connectivity with category-sensitive regions of the neocortex, which I think is fine. They note also that this choice is exploratory. But, they in their motivation do not really touch on the fact that integration itself has generally been more ascribed to the anterior hippocampus-so it seems to me that there is a possibility that either region could be important, and if the authors want they could perhaps set those dual possibilities up. Perhaps replay via the posterior hippocampus of the single events with later integration in the neocortex is one mechanism, but online integration & replay of those associations (perhaps more of an anterior hippocampus operation) is another. It then perhaps becomes clearer why it is interesting that the authors find the effect specific to the posterior hippocampus, with nothing much going on in the anterior. Could the work speak to these two possibilities (i.e., the findings seem more consistent with the first possibility of immediate storage for the memories separately, and offline integration via reactivation of the constituent traces)? Alternatively, might the authors explain why they think integration per se in the anterior hippocampus seen in previous work should not be related to the post-encoding connectivity profile?

We agree that the specific roles of anterior and posterior hippocampus is an interesting area of focus that could be better discussed in the context of our experiment. First, we provided some additional justification for focusing on the posterior hippocampus based on category-selectivity of the encoded stimuli in the Introduction (page 7, lines 139-159). Then, in the Discussion, we have provided additional background and some speculation of the different roles that the two regions may be playing in the formation and offline processing of specific memories and their integration (pages 36-37, lines 978-986).

(3) p. 7, line 142 – It was not clear to me why RT (per se, or in all cases) is an ideal readout of memory integration. Can the authors provide more explanation on this point?

Thank you for pointing out the need to clarify this point. In brief, we chose to use priming and RTs as measure of integration because we reasoned that it would minimize more goal-oriented, on-the-fly retrieval strategies that might be used during standard explicit tests of integration. There is ample evidence that response time measures reflect various aspects of memory integration, including in tests of lexical integration, statistical learning, fast mapping, and more. We have now included this logic in the introduction (page 8, lines 167-173).

(4) It was unclear from the Introduction what the reason was for the separate behavioral experiment. Up until p. 13, it seems as though Experiment 2 basically serves to replicate & extend Experiment 1, and only later is it clear that there are interesting nuanced differences in the explicit test. Can this be stated earlier (in the Intro or earlier in the Results)?

We did indeed intend for Experiment 2 to be a replication of Experiment 1, with the added opportunity for investigating neural signatures of consolidation as they relate to the priming effect. Another point that we clarified is that we have always considered the priming task to be the main task of interest; the explicit memory tests were added for validation and comparison to past integration work which mostly employs explicit memory tests. This is also why the explicit task changed across experiments; it was intended as a secondary, more exploratory measure. We have clarified this logic in the introduction (page 8, lines 191-196).

(5) p. 9 – For the behavioral results (RT) for the sequence learning task, please clarify whether only correct responses were included in this model. Or, if all trials, it would be helpful to clarify why the decision to include incorrect trials as well.

We used all trials in this analysis. We went this approach because we arranged the sequences that the objects according to their sizes, so motor responses would be matched across conditions (e.g. responses for B-C1 and B-C2 from overlapping sequences could be smaller-bigger, while for another set of sequences, it could be bigger-smaller, and these would be matched across recent and remote learning). However, because we pseudo-randomly intermixed the order of sequences and baseline objects, a large portion of the comparisons participants made would be between two big objects or two small objects, and in these cases, there may not be an objectively correct answer, meaning we were not able to identify whether participants answered any incorrectly. For this reason, we included all trials in this analysis regardless of accuracy. We clarified this in the Methods section (page 44, lines 1173-1186).

(6) It took me a while to work out exactly how the priming task was set up, and I think more details in the main paper would have been useful. It was unclear to me until I read the Methods that every sequence had one C item that appeared in the "primed" condition and one in the "control" condition (and that, as noted above, the control transition always has to happen first, by definition). I think it would be helpful to be more specific in the main paper about the critical comparison-specifically, more on the control condition. On first read, I was not sure whether C trials preceded by novel foils were used as a control (which would not be ideal, but I don't think this is what the authors did). I think baseline objects were old objects and were presented an equal number of times as the C items (based on p. 35); putting this in the main text would have been so helpful. It might also be helpful to note that each object appeared just once in the priming task, to help readers get a sense of how many trials/sequences (or the max #) go into each comparison.

Thank you for highlighting our need to clarify the priming task. You are correct that the control condition used baseline items as they were presented an equal number of times as the C objects and, if participants have intact recognition memory, their responses to all items in a priming sequence (baseline, C1, C2) would be ‘old’, thus preventing lags in response times due to a change in the response decision and motor movement. We have clarified this in the main text (page 13, lines 307-311).

(7) I was a bit confused about the use of the words "baseline" and "control" objects (e.g., on p. 11)-it seemed at times the terms were being used interchangeably, and in other places, the baseline objects were what made up the control condition.

Thank you for catching this. We intend for the word ‘baseline’ to represent the objects presented during encoding that did not belong to any sequence, and for the control condition to reflect performance on the C objects preceded by baseline objects during the recognition priming task. We have edited these terms throughout the manuscript to be consistent with these two definitions. We also changed all mentions of ‘items’ to ‘objects’ for more consistency in terminology.

(8) p. 13 – at the end of the section on priming, possibly worth qualifying the final sentence to note the effect of order as being present (interestingly) only for the remote sequences?

Thank you for this suggestion – we have modified that section to highlight that our findings are only for the remote sequences (now on page 15 line 365).

(9) p. 13, line 317 – where it says accuracy was reliably above chance, I think that means there is significant evidence for integration (more familiarity for intact vs. rearranged pairs), but I wasn't totally sure since it wasn't called accuracy earlier. Please clarify.

We see have modified this section to use the term behavioral integration rather than accuracy, which makes for more consistent language across the two experiments (pages 15-16, lines 381-382 and 387-389).

10. For p. 15 – "all object images" presented during the pre- and post-learning exposure phases: I assume this is all objects from the remote and recent learning tasks, not the novel foils from the priming task.

Correct – we have clarified this (Page 17, lines 421-422).

(11) p. 17 – I was confused about this partial correlation control analysis. Did this analysis use a similar approach to the main analysis, where the authors took some kind of average score per participant? (Would that be average across the matrix diagonal, i.e., between all pairs of related C objects?) Or, was this somehow used to control for differences in BOLD signal between C1 and C2 in a trial-by-trial way?

Thank you for highlighting our need to clarify this section. This was done on a pair-by-pair manner, by creating a matrix with the same order of C1 and C2 objects with cells that capture the difference in average BOLD signal between each pair of objects. Then this matrix was re-arranged into a vector (in the same manner as the elements of the neural integration model and the similarity matrix for a given ROI and session) and entered into partial correlation analyses. The output of this analysis is a measure of fit for each participant, between neural data and the model, while accounting for BOLD signal. These fits could then be analyzed in the same manner as the main analyses.

We have taken steps to clarify this analysis in a few ways. First, we added a paragraph in the Methods section that described this analysis in more detail (pages 52-53, lines 1402-1414). Second, we summarized and clarified the conceptual aim of the analysis in the Results section (page 20, lines 505-512).

(12) p. 38 – "Accuracy computed for any learning trial in which the preceding object was coded to be a different size (e.g. an object bigger than a shoebox following an object smaller than a shoebox), as comparisons amongst objects that were both bigger or both smaller are likely more variable and dependent on participants' subjective opinion." line 987-990 – is there a portion (clause? verb?) of this sentence missing?

This was a typo and has been fixed: “Accuracy *was* computed for…”

(13) p. 39 – what instructions were given prior to the explicit memory test, especially on how to do the C1-C2 association trials?

The instructions for this test were very explicit: “You may have noticed that during the size task, some objects often appeared in sequences. For instance, maybe two objects always appeared back-to-back, Or, maybe two objects were always preceded by the same sequence.” This was accompanied by a visualization of a pair of overlapping ABC1 and ABC2 sequences. They were then instructed to perform the task based on how familiar each pair of objects felt specifically based on the sequence information they learned during learning. We have included this extra information to the Methods section (page 46, lines 1230-1235 for Expt 1 and pages 49-50, lines 1316-1317 for Expt 2).

(14) I thought it was interesting that the authors decided to match the C1 and C2-related objects in terms of size judgment so that the motor history was the same. I was curious as to whether the authors thought this might have encouraged integration since both objects point toward the same response (vs. a case where differentiating the two objects might be more important, as when the response depends on object identity). Just a thought!

We did think about motor history as a possible contributor to integration or differentiation, which is why we decided to match the size judgments for C1 and C2 objects from overlapping sequences. We don’t have concrete evidence that motor history would impact the ability to integrate across sequences, but we decided on this path to reduce variability and potential sources of differentiation. We would be interested in testing this systematically in the future!

(15) The authors were generally very thorough in their references to the literature, but they might want to consider citing these additional papers that are relevant: Abolghasem et al. (2023) Developmental Science; Ye et al. (2020) eLife. Also, since the Audrain and colleagues' paper is now published, please be sure to switch the citation from the biorxiv.

Thank you for bringing these relevant papers to our attention – we have updated our manuscript accordingly in both the Introduction and Discussion, and we also incorporated several papers that have been published since our original submission.

Reviewer #2 (Recommendations for the authors):(1) A table outlining transition probabilities between all relevant item types in the sequence learning and the implicit recognition priming task would be helpful.

We have added a table of transition probabilities across the learning, recognition priming, and exposure scans (now Table 1, page 48).

(2) Also relating to the public review, a clear line should be drawn between planned and exploratory analyses, and multiple comparison corrections should be applied to exploratory analyses that perform the same contrast on multiple ROIs.

We have now specified in each analysis section which ROI was planned and which contrasts were exploratory, and we have corrected for multiple comparisons across all regions included in each analysis.

(3) Statistics should be reported for the analysis of z-scoring across voxels to control for the influence of the overall BOLD signal on the neural pattern analysis.

This is a common pre-processing step for multi-variate pattern analysis approaches that down-weights the contribution of noisy voxels to patterns of BOLD signal that correspond to each stimulus (Kuhl & Chun, 2014: Richter et al., 2016). We have edited the Results section to clarify the purpose of this statistical transformation, confirm that the results do not meaningfully change, and cite other work that has employed this step (pages 19-20, lines 501-502).

(4) For the partial correlation control analysis, the methods and results should be clear about whether the BOLD differences between C1 and C2 represent pre- or post-learning distances (or both)?

They were computed separately for pre- and post-learning exposures. We have clarified this analysis accordingly, both in the Results and Methods section (see R1.11 for more details).

(5) Work by the Kuhl lab on the repulsion of overlapping memories (including spatial sequence memories) appears highly relevant and should be discussed.

We have integrated the lab’s work into our discussion in two ways: first, by discussing potential reasons for differences in how overlapping memories are processed (either differentiation or integration) and second, by highlighting their observations of neural differentiation in the hippocampus, rather than what we observe in the mPFC. The discussion is much more nuanced as a result (page 33, lines 865-868; page 32, lines 847-854).

(6) The sequence learning (i.e., size judgment) task puts a strong emphasis on the temporal relationships between items, and the priming task has a strong explicit recognition memory component. Could the authors briefly comment on whether they believe their results generalize to entirely implicit settings with no such explicit temporal binding or explicit memory demands?

This is a great open question. Our intuition is that the results would also generalize to an implicit setting, but without an explicit temporal binding, their ability to learn the sequences would take longer. In early piloting for this experiment and other experiments in the lab that examine temporal binding, it seems clear that some attention to the temporal structure is needed for the sequential information to be learned in one experimental session of a reasonable length. For example, in very early stimulus and task development for this experiment, a handful of participants were exposed to the sequences while completing a size judgment task for each image (bigger or smaller than a shoebox). Here, there was no evidence of priming, but there was also no evidence of learning ABC_1_ and ABC_2_ based on participants’ response times for the size judgments.

In terms of an explicit memory demand, we predict that our findings would generalize to other tasks easily. Indeed, similar measures of integration have been observed using preference ratings (Abolghasem et al., 2023), and it seems reasonable to expect that a task like a size judgment or a similar such judgment would also work. We chose a recognition task because we wanted motor responses for each item in the sequence to be matched to avoid any variability in responses due to participants changing their response across priming trials (Dobbins et al., 2004). To this end, participants responded ‘old’ to all items that were included in those analyses. However, variability due to motor responses could be accounted for in additional ways with these other tasks.

Reviewer #3 (Recommendations for the authors):(1) The validity of the observed behavioral priming effect could be more rigorously tested to improve the interpretation. Currently, the authors compare the reaction times of target C2 preceded by C1 from the overlapping sequence (prime condition), with a condition where C1 is preceded by a baseline object that is not a part of any sequence (control condition). Do they still see a priming effect when C1 is preceded by a C object from a non-overlapping sequence instead of the baseline objects (is this even possible with the task design)? This would test the specificity of the priming effect. If C objects are integrated due to the overlapping components of their sequences, one should observe priming in this condition but not when preceded by C objects from non-overlapping conditions. Alternatively, C objects could all be integrated with each other due to some other dimension of the task, such as the fact that C objects are presented less frequently than A or B or are less predictable in general than B, etc.

We did not include a control condition in which C objects from different sequences appeared back-to-back, as we wished to maximize the number of comparisons in the primed condition for power purposes while only presenting each object once. Repetition of objects during the priming task may have further facilitated response times and potentially masked any priming effects. We agree that this omission leaves open the possibility that the facilitated responses to C2 may be due to an integration of all C objects due to their shared transition probabilities, and their ‘place’ as the third items in triplet sequences. However, it is encouraging that the observed priming effect for the remotely learned sequences is selectively correlated with post-learning neural similarity between C objects from overlapping sequences, and not related to similarity between C objects from different sequences (Supplemental Figure 4). It’s also worth noting that the explicit integration tests were always conducted with C objects from other sequences as foils; above-chance performance in both experiments suggests that participants successfully learned sequence-specific relationships rather than information about objects’ positions in their sequences. Regardless, we have included this outstanding question as a caveat in the Discussion (pages 40-41, lines 1087-1107).

(2) It would be helpful to get a sense of the overall model fits for the neural data to the integration model in the mPFC and LOC, as we only see the change in fit, making it difficult to understand how well these models fit the data in the first place. In addition, could a decrease in fit be due to integration occurring across C items from non-overlapping sequences, instead of differentiation of C items from overlapping sequences?

It is clear from multiple reviewers that these data would be helpful to include. We have included these model fits along with appropriate statistical tests. Please see Reviewer #1 comment #2 for more details, but as a reminder, the decrease in model fit for mPFC for the remote sequences was driven primarily by a decrease in similarity for the overlapping C items and not the non-overlapping ones. Interestingly, in LOC, all C items grew more similar after learning, regardless of their overlap or learning session, but the increase in model fit for C items in the recent condition was driven by a larger increase in similarity for overlapping pairs relative to non-overlapping ones.

(3) For the neural measures related to the priming section (page 23), I was confused as to why the authors took a model comparison approach to determine the unique variance explained by each predictor. Isn't it the case that when you include both similarity and connectivity in the model, if both remain significant predictors, they explain unique variance in priming? You could then calculate the semi-partial R2 to determine exactly how much unique variance is explained by each. I point this out because model comparisons are anti-conservative given that a model with more predictors will inherently explain more variance than a model with fewer predictors.

This makes sense to us and we have changed our statistical approach accordingly. Please see page 27, lines 706-710, for updated statistics and page 55, lines 1478-1482, for Methods; the results have not changed.

(4) The authors frequently refer to their connectivity results as reflecting post-encoding reactivation. I understand that there is evidence from other studies that reactivation occurs during the post-encoding period, but the authors have not shown this (I believe one would need to see patterns evident of encoding being reinstated during the post-encoding period). It would be more accurate to refer to the results in terms of post-encoding coupling rather than reactivation throughout, so as not to over-interpret in light of the data.

We have modified our terminology to use ‘post-encoding coupling’ rather than ‘reactivation’ throughout the manuscript, as it is not a direct measure of reactivation of specific memories. We also, however, clarify our use of hippocampal-cortical coupling as one of several indexes of post-learning reactivation in the Introduction (page 5, lines 95-101). While hippocampal-cortical coupling cannot capture stimulus-specific reactivation, it is selective to category-selective cortical regions (de Voogd et al. 2016; Murty et al. 2017) and selective post-learning rest periods rather than pre-learning indices of coupling, making it a likely signature of coordinated reactivation across hippocampus and cortex.